

# StreamFlow 1.0: An extension to the spatially distributed snow model Alpine3D for hydrological modeling and deterministic stream temperature prediction

Aurélien Gallice[1], Mathias Bavay[2], Tristan Brauchli[1], Francesco Comola[1], Michael Lehning[1,2], and Hendrik Huwald[1]

[1]School of Architecture, Civil and Environmental Engineering (ENAC), École Polytechnique Fédérale de Lausanne (EPFL), Switzerland
[2]SLF, WSL Institute for Snow and Avalanche Research, 7260 Davos, Switzerland

*Correspondence to:* M. Bavay (bavay@slf.ch)

**Abstract.** Climate change is expected to strongly impact the hydrological and thermal regimes of Alpine rivers within the coming decades. In this context, the development of hydrological models accounting for the specific dynamics of Alpine catchments appears as a one of the promising approaches to reduce our uncertainty on future mountain hydrology. This paper describes the improvements brought to *StreamFlow*, an existing model for hydrological and stream temperature prediction built as an

5   external extension to the physically-based snow model *Alpine3D*. *StreamFlow*'s source code has been entirely written anew, taking advantage of object-oriented programming to significantly improve its structure and ease the implementation of future developments. The source code is now publicly available online, along with a complete documentation. A special emphasis has been put on modularity during the re-implementation of *StreamFlow*, so that many model aspects can be represented using different alternatives. For example, several options are now available to model the advection of water within the stream. This

10   allows for an easy and fast comparison between different approaches and helps in defining more reliable uncertainty estimates of the model forecasts. In particular, a case study in a Swiss Alpine catchment reveals that the stream temperature predictions are particularly sensitive to the approach used to model the temperature of subsurface runoff, a fact which has been poorly reported in the literature to date. Based on the case study, *StreamFlow* is shown to reproduce hourly mean discharge with a Nash-Sutcliffe efficiency (NSE) of 0.82, and hourly mean temperature with a NSE of 0.78.

## 1   Introduction

Mountainous areas play a major role in hydrology by accumulating precipitation as snow and ice during the winter and redistributing it as melt water during spring and summer. Downstream areas hereby receive larger amounts of water during the hot season, when demand – especially in terms of agriculture – is highest. In fact, Viviroli et al. (2011) estimate that more than 40% of the world's mountainous regions provide an important supply for low-land water use. Accordingly, more than

20   one sixth of the world's population is currently living in areas depending on snow melt for their water supply (Barnett et al., 2005). Apart from its relevance for downstream areas, mountain hydrology also strongly impacts hydropower production (e.g.



Schaefli et al., 2007; Finger et al., 2012; Majone et al., 2016), determines the habitat suitability of numerous aquatic organisms (e.g. Short and Ward, 1980; Hari et al., 2006; Wilhelm et al., 2015; Padilla et al., 2015) and even plays a noticeable role in the global emission of carbon dioxide into the atmosphere (Butman and Raymond, 2011; Raymond et al., 2013).

Mountainous environments have recently been identified as being especially sensitive to climate change (Barnett et al., 2005; Stewart et al., 2005; Viviroli et al., 2011, e.g.). In particular, winter air temperature over the last 70 years has been observed to increase by more than twice the global mean in the European Alps (Beniston, 2012), and this trend is forecasted to remain unchanged in the next decades (Kormann et al., 2015b). Rising air temperature will be responsible for less precipitation falling as snow in winter and an earlier onset of snow melt in spring (Barnett et al., 2005; Bavay et al., 2009, 2013, e.g.). As a consequence, the spring freshet will occur earlier in the season and, assuming mean annual precipitation to remain constant, will also have a reduced magnitude (e.g. Stewart et al., 2005; Kormann et al., 2015b, a, to name just a few). Some studies predict an increase in winter precipitation, which could at least partially compensate for the decreased fraction of solid precipitation and sustain the spring freshet close to its actual level (Schaefli et al., 2007; Beniston, 2012; Finger et al., 2012; Fatichi et al., 2015). Autumn and winter stream discharge is expected to increase in magnitude and variability as a result of the higher fraction of precipitation falling as rain, which might result in greater flood risks in winter (Barnett et al., 2005; Bavay et al., 2009; Finger et al., 2012; Beniston, 2012). Summer discharge will likely be much reduced and the drought risks therefore more pronounced, at least in the watersheds with little or no glacier cover (Schaefli et al., 2007; Stewart et al., 2015). In glaciated catchments, increased summer ice melt might (over)compensate for the reduced snow melt on an annual average basis (Bavay et al., 2013; Kormann et al., 2015a). This compensation is however expected to last only until the glaciers have shrunk to the point where ice melt discharge starts to decrease as well, a phenomenon which has already been observed in some parts of the world (see e.g. studies mentioned in Kormann et al., 2015a). In summary, the hydrological regimes of many mountainous catchments are forecasted to shift from glacio-nival and nival signatures to nivo-pluvial or even pluvial regimes (Aschwanden and Weingartner, 1985; Beniston, 2012).

As a result of the changes in climate and hydrological regime, the thermal regime of the mountain streams will change as well in the coming decades (e.g. Morrison et al., 2002; Null et al., 2013; Ficklin et al., 2014; Stewart et al., 2015). Due to the strong correlation between stream and air temperatures (e.g. Mohseni et al., 1998; Caissie, 2006), the increase in air temperature is expected to result in globally higher stream temperatures over the year (e.g. Ferrari et al., 2007; Ficklin et al., 2012). The increase in mean annual precipitation predicted by some studies will only slightly mitigate this temperature rise through an increase of the mean annual discharge – and hence the heat capacity – of the streams (Ficklin et al., 2012, 2014). The reduction of the spring freshet will diminish the buffering effect of snowmelt on stream temperature, hereby leading to larger stream temperature increases in spring (Ficklin et al., 2014). Similarly, lower summer flows in little glaciated catchments are likely to result in increased mean summer stream temperature and more frequent extreme temperature events (Stewart et al., 2005; Null et al., 2013). All these predictions support the hypothesis that stream temperature will respond in a non-linear way to the air temperature rise.

The climate change induced modifications of the hydrological and thermal regimes of alpine streams are expected to strongly impact their ecology. The forthcoming air temperature rise will lead to a modification of the riparian vegetation, which in turn



will affect the stream ecosystem (Hauer et al., 1997). The higher stream temperatures will also have consequences on the cold water fish species encountered in mountain streams, whose fry emergence date (Elliott and Elliott, 2010), growth rate (Hari et al., 2006) and death rate (Wehrly et al., 2007) are all mostly dependent on stream temperature. Future increases in stream temperature are expected to result in a shift of the suitable habitat for such species to higher elevations, where dams

and other physical barriers might limit their migration and imply a reduction of their habitat (Hauer et al., 1997; Hari et al., 2006). Padilla et al. (2015) report that the summer stream discharge variability is currently increasing, which is detrimental to the spawning rate of the fishes. However, they note that reduced spring discharge might partly compensate for the increase in stream temperature by facilitating the upstream migration of the fishes.

Hydropower production might also suffer from the effects of climate change on alpine hydrology (e.g. Schaefli et al., 2007;

Beniston, 2012; Fatichi et al., 2015). This fact is all the more worrying in the current context of transition towards renewable energy sources, especially for small alpine countries such as Switzerland which heavily rely on hydropower for their electricity production (Schaefli, 2015). Several studies point at the future decrease of up to 36% in the energy production of the dams located at high altitudes (Schaefli et al., 2007; Finger et al., 2012; Fatichi et al., 2015), resulting from the shift of the hydrological regime from glacio-nival to pluvial-nival. Schaefli et al. (2007) also mentions that the spillway – an emergency van intended to

avoid dam overflow – may have to be occasionally activated in the future, with all the dramatic consequences that it entails for downstream areas.

The modification of the stream ecology and the reduction of the hydropower production are only two examples of the consequences of climate change on mountain streams. In order to better evaluate and predict these consequences, numerous numerical models have been developed over the last decades. Most of them concentrate either on the prediction of discharge

(e.g. Grillakis et al., 2010; Bürger et al., 2011; Schaefli et al., 2014; Ragettli et al., 2014) or water temperature (e.g. Caldwell et al., 2013; Tung et al., 2014; Hébert et al., 2015; Toffolon and Piccolroaz, 2015), but few are able to simulate the two at the same time (e.g. Loinaz et al., 2013; MacDonald et al., 2014; Comola et al., 2015). Regarding the models predicting only discharge, they can be classified – among other possibilities and in order of increasing spatial resolution – either as lumped, semi-distributed or fully distributed (e.g. Khakbaz et al., 2012). Lumped models are often based on empirical equations and only

allow for the computation of stream discharge at the catchment outlet. Fully distributed models, on the other hand, typically solve the full mass and momentum conservation equations, but require extensive computational resources (e.g. Beven, 2012). As a trade-off between the two approaches, semi-distributed models have become quite popular over the last decades, since they can be applied over large areas while at the same time accounting for sub-catchment characteristics (Khakbaz et al., 2012; Beven, 2012). An equivalent sort of classification is commonly applied to stream temperature models, which are usually

separated into statistical and mechanistic models (Caissie, 2006). Statistical models require less input data and are usually easier to use, but their lack of physical basis is often seen as a limit to the validity of their predictions in the context of climate change studies (e.g. Piccolroaz et al., 2016). On the contrary, more credit is generally given to the long-term forecasts of the deterministic stream temperature models, although their accuracy is about the same – if not worse (Ficklin et al., 2014) – than the statistical models. It should be mentioned that an intermediate sort of model, referred to as hybrid, has recently been



developed (Gallice et al., 2015; Toffolon and Piccolroaz, 2015) and shown by Piccolroaz et al. (2016) to be suitable for climate change studies.

As opposed to the separate simulation of discharge and stream temperature, the coupled modeling of the two offers new perspectives to investigate the effects of climate change on mountain hydrology (e.g. Ficklin et al., 2014). For example, the

variations of temperature resulting from the fluctuations in discharge can be better resolved (e.g. van Vliet et al., 2012; Null et al., 2013). The use of both discharge and temperature measurement data to calibrate the model has also been shown by Comola et al. (2015) to improve the quality of the simulation. Surprisingly, only a few coupled hydro-thermal models have been developed to date (see Table 1), probably as a result of the rather small size of the scientific community involved in stream temperature research. Out of the 13 semi-distributed coupled models listed in Table 1, only one was specifically developed for

mountainous environments (MacDonald et al., 2014). The other ones were either tailored to large-scale applications (Morrison et al., 2002; Ferrari et al., 2007; van Vliet et al., 2012; van Beek et al., 2012; Null et al., 2013) or aimed at being used over low-altitude catchments (e.g. Sullivan et al., 1990; Chen et al., 1998; Haag and Luce, 2008; Sun et al., 2015). In addition, all of these models simulate the snowpack energy-balance using a more or less simplified approach, most of them relying on the degree-day method (e.g. van Beek et al., 2012; Null et al., 2013; MacDonald et al., 2014).

The present study aims at presenting the improvements brought to the semi-distributed model recently developed by Comola et al. (2015) for coupled streamflow discharge and temperature simulations. This model, referred to as *StreamFlow* in the following, was specifically developed for high Alpine environments as it builds upon the detailed snow model *Alpine3D* (Lehning et al., 2006). It was decided to entirely rewrite the code of Comola et al. so as to fully exploit the advantages offered by object-oriented programming in terms of flexibility and code structure. In particular, the new model is much more

modular, allowing for various components of the hydrological cycle to be modeled using different approaches. Some of these approaches have been implemented which were not present in the original model of Comola et al., hereby offering a wider range of modeling possibilities to the end user. The mass- and energy-balance equations implemented in the model are detailed in Sect. 2, and the new code structure in Sect. 3. The model is applied to a case study in Sect. 4 in order to demonstrate some of its features and provide an assessment of its accuracy. Conclusions are found in Sect. 5.

## 25   2    Model description

*StreamFlow* is built as an independent extension to the spatially-distributed snow model *Alpine3D* (Lehning et al., 2006, 2008). The latter was developed to study multiple subjects such as the impact of climate change on snow cover (Bavay et al., 2009, 2013), the effect of wind and topography on snow deposition (Mott and Lehning, 2010; Mott et al., 2014) or the sublimation of drifting snow (Groot Zwaaftink et al., 2013). *Alpine3D* operates on a regular mesh grid, and essentially runs the one-

dimensional *Snowpack* model over each grid cell independently. *Snowpack* computes the time evolution of the vertical snow profile, as well as the vertical profiles of soil moisture and soil temperature (Bartelt and Lehning, 2002; Lehning et al., 2002b, a). It accounts for the canopy layer (Gouttevin et al., 2015) and can simulate the vertical water transport using either the Richards equation or a simple bucket scheme (Wever et al., 2014, 2015).





*StreamFlow* is implemented as a semi-distributed model, i.e. based on the subdivision of the catchment into subwatersheds. This subdivision is typically performed using the well-known tool suite TauDEM (Tarboton, 1997), which extracts both the stream network and its corresponding set of subwatersheds from the digital elevation model (DEM). The stream network is automatically partitioned into so-called *stream reaches*, where each reach is uniquely associated with a subwatershed and

corresponds to the portion of the stream network which specifically drains the subwatershed in question. It should be stressed out that subwatersheds are independent and distinct from each other, i.e. they do not spatially overlap and are considered not to interact from a hydrological point of view. Stream reaches, on the other hand, are connected to each other: the computation of discharge and temperature in a given reach requires the same variables to be computed in its upstream tributaries first.

As schematically represented in Fig. 1, *StreamFlow* pursues the simulation of the water flow from the point where *Alpine3D*

stops to model it. Each subwatershed is approximated in *StreamFlow* as a linear reservoir. The total percolation rate computed by *Alpine3D* at the bottom of all the soil columns belonging to a given subwatershed is considered by *StreamFlow* as the inflow rate into the associated linear reservoir. The latter then computes the discharge and temperature of the subsurface runoff flowing out of the subwatershed. Note that the term *subsurface runoff* will be used in the remaining of this paper as a generic word standing for both the fast and slow components of the subsurface runoff, which are sometimes referred to as interflow

and baseflow in the literature. Subsurface runoff produced by each subwatershed is delivered as lateral inflow to its associated stream reach (see Fig. 1). In other words, the subwatersheds are used in *StreamFlow* to compute the amount of subsurface water and heat penetrating the stream network. As such, the model is only able to reproduce so-called *gaining streams*, as opposed to *loosing streams* which would require a mechanism to transfer water from the stream network to the subwatersheds. As a final step, *StreamFlow* advects water and energy within the stream network down to the catchment outlet point. To this end,

discharge and temperature are computed within each stream reach, taking notably the water and heat inflows originating from the upstream reaches and from the subsurface runoff into account. The different processing steps of *StreamFlow* are described into more detail below.

## 2.1   Subwatershed modeling

In *StreamFlow*, the discharge $Q_{\mathrm{subw}}$ $(\mathrm{m^3\,s^{-1}})$ of subsurface runoff is computed independently from its temperature $T_{\mathrm{subw}}$ (K).

This allows for the different temperature modeling approaches to be combined with every discharge computation alternative.

### 2.1.1   Water transfer

Only the linear reservoir approach developed by Comola et al. (2015) has been implemented so far for the estimation of the subsurface runoff discharge, but the modular structure of *StreamFlow* supports the integration of more complex, physically-based algorithms. The approach of Comola et al. represents each subwatershed as two superposed linear reservoirs, the lower

one being filled at a maximum inflow rate $R_{\mathrm{max}}$ $(\mathrm{m\,s^{-1}})$ and the upper one receiving the excess inflow water. The model behavior is controlled by three user-specified parameters: the mean characteristic residence times $\overline{\tau}_{\mathrm{res,u}}$ (s) and $\overline{\tau}_{\mathrm{res,l}}$ (s) in the upper and lower reservoirs, and $R_{\mathrm{max}}$. The complete mathematical background underlying this approach is detailed in Comola et al. (2015); a summary of the main equations and an explanatory figure can be found in Appendix A. Depending on the



approach used to spatially discretize the stream reaches, water flowing out of each subwatershed is either transferred to its associated reach as a whole or partitioned between the different cells composing the stream reach (see below).

### 2.1.2 Computation of the subwatershed outflow temperature

Three alternatives are available in *StreamFlow* for the estimation of subsurface runoff temperature. The first approach corresponds to the one developed by Comola et al. (2015), which requires subsurface runoff to be modeled as in Sect. 2.1.1 above and is therefore not compatible with potential future alternatives for subsurface runoff discharge modeling. It performs a simplified energy-balance of subsurface water at the subwatershed scale. The temperature of water stored in each one of the two superposed reservoirs is computed based on the temperature of infiltrating water, taking thermal exchange with the surrounding soil into account. This model requires the specification of a parameter, $k_{soil}$ (s), which corresponds to the characteristic time of thermal diffusion between the water stored in the reservoirs and the soil. The complete description of this technique can be found in Comola et al. (2015) and is also summarized in Appendix A for convenience.

The second method implemented in *StreamFlow* for the computation of $T_{subw}$ is adapted from the approach used in the Hydrological Simulation Program–Fortran (HSPF Bicknell et al., 1997). This technique essentially approximates the time evolution of $T_{subw}$ by smoothing and adding an offset to the time series of air temperature $T_a$ (K),

$$\frac{dT_{subw}}{dt} = \frac{1}{\tau_{HSPF}}\left(T_a - T_{subw} + D_{HSPF}\right). \tag{1}$$

In the above equation, $T_a$ is taken as the mean air temperature over the subwatershed as computed by *Alpine3D*, and the smoothing coefficient $\tau_{HSPF}$ (s) and the temperature offset $D_{HSPF}$ (K) can be freely specified by the user. This equation is solved in *StreamFlow* using a second order Crank-Nicholson scheme.

Finally, the third technique for estimating the temperature of subsurface flow relies on the assumption that infiltrated water is in thermal equilibrium with the surrounding soil matrix. As such, $T_{subw}$ can be considered to have the same value as the local soil temperature $T_{soil}$ averaged between the soil surface and a given depth $z_d$ (m). In practice, $T_{subw}$ is determined at any point along the stream network by identifying the cell of the *Alpine3D* mesh in which it is located, and then averaging the soil temperature values computed by *Alpine3D* in this cell down to depth $z_d$.

### 2.2 Stream network modeling

As mentioned above, the computation of discharge and temperature within the stream network is based on the subdivision of the latter into reaches. Each reach is uniquely associated with its corresponding subwatershed and is automatically identified by TauDEM based on a geomorphological analysis of the DEM. The stream reaches can be modeled in *StreamFlow* using two different approaches (see Fig. 2):

(a) A lumped approach, in which each reach is treated as a single entity whose mean water depth, outlet discharge and temperature are to be computed. This method was already implemented by Comola et al. (2015) in the first version of *StreamFlow*. In this approach, each reach collects the subsurface runoff originating from its associated subwatershed as a whole – no spatial discretization is performed.





(b) A discretized approach, which subdivides each reach into smaller spatial units referred to as *cells* in the following. The cells are delineated using the grid pattern of the DEM used by TauDEM to identify the subwatersheds and the stream network (see Fig. 2); as a consequence, all cells do not have the same length within a single reach. This discretization method provides higher spatial resolution than the lumped approach and supports more advanced techniques for water and temperature routing (e.g. the resolution of the shallow water equations). In this approach, the water flowing out of each subwatershed is transferred to the cells of its corresponding stream reach, proportionally to the specific drainage area of each cell.

The different methods available in *StreamFlow* for in-stream routing of water and energy are described below.

### 2.2.1 Water routing

Stream discharge can be computed using two different approaches, which can both be used with lumped or discretized reaches. A third approach, namely the shallow water equation solver for the discretized reaches, is currently being developed and should be available in the near future.

The first water routing technique is the same as the one already available in the original version of *StreamFlow*, namely the instantaneous advection of water down to the catchment outlet. This approach is based on the fact that, in small catchments, the amount of time required for a rain drop to reach the catchment outlet is mostly dominated by the time spent within the hillslopes (see e.g. Comola et al., 2015). Water depth $h$ (m) is computed using a power function of discharge $Q$ ($\mathrm{m^3\,s^{-1}}$), i.e. $h = \alpha_h Q^{\beta_h}$, where the coefficients $\alpha_h$ and $\beta_h$ can be calibrated or specified a priori.

The second approach corresponds to the well-known Muskingum-Cunge technique, shown by Cunge (1969) to be a diffusive-wave approximation of the shallow water equations. *StreamFlow* implements the modified three-point variable parameter method developed by Ponce and Changanti (1994), which is first-order accurate in time and second-order in space. This method can be used with both lumped and discretized stream reaches. In discretized reaches, it estimates discharge $Q_i^{n+1}$ ($\mathrm{m^3\,s^{-1}}$) at the outlet of cell $i$ at time $t_{n+1} = t_n + \Delta t$ as (see e.g. Tang et al., 1999):

$$Q_i^{n+1} = c_1 Q_{i-1}^n + c_2 Q_{i-1}^{n+1} + c_3 Q_i^n, \tag{2}$$

where $\Delta t$ (s) denotes the time step, $Q_{i-1}^n$ the sum of the outlet discharge of cell $i-1$ and the lateral subsurface flow discharge into cell $i$ at time $t_n$, and the coefficients $\{c_k\}_{k=1,2,3}$ (–) are computed as:

$$c_1 = \frac{k_i x_i + 0.5\Delta t}{k_i(1-x_i) + 0.5\Delta t},$$
$$c_2 = \frac{-k_i x_i + 0.5\Delta t}{k_i(1-x_i) + 0.5\Delta t},$$
$$c_3 = \frac{k_i(1-x_i) - 0.5\Delta t}{k_i(1-x_i) + 0.5\Delta t}.$$



Parameters $k_i$ (s) and $x_i$ (–) can be related to hydraulic properties of the stream cell,

$$k_i = \frac{l_i}{c_\mathrm{r}}, \tag{3}$$

$$x_i = \frac{1}{2} \min\left(1, 1 - \frac{Q_\mathrm{r}}{c_\mathrm{r} w S_0 l_i}\right), \tag{4}$$

with $l_i$ (m) denoting the cell length, $w$ (m) the stream width, $S_0$ (–) the local bed slope in cell $i$, $c_\mathrm{r}$ ($\mathrm{m\,s^{-1}}$) a representative

wave celerity and $Q_\mathrm{r}$ ($\mathrm{m^3\,s^{-1}}$) a representative discharge. Manning's formula is used to derive $c_\mathrm{r}$ from $Q_\mathrm{r}$ under the assumption

of a rectangular channel cross-section,

$$c_\mathrm{r} = \frac{5}{3}\left(\frac{S_0}{n_\mathrm{m}^2}\right)^{3/10}\left(\frac{Q_\mathrm{r}}{w}\right)^{2/5}, \tag{5}$$

where $n_\mathrm{m}$ ($\mathrm{s\,m^{-1/3}}$) is the Manning coefficient, whose value is generally accepted to be within the approximate range 0.03–0.10

for small natural streams (e.g. Phillips and Tadayon, 2006). $Q_\mathrm{r}$ is computed as:

$$Q_\mathrm{r} = \frac{Q_{i-1}^n + Q_{i-1}^{n+1} + Q_i^n}{3}. \tag{6}$$

Manning's formula is also used to determine the water depth $h_i^{n+1}$ (m) in cell $i$ at time $t_{n+1}$ based on $Q_i^{n+1}$:

$$h_i^{n+1} = \left(\frac{n_\mathrm{m} Q_i^{n+1}}{w S_0}\right)^{3/5}. \tag{7}$$

In order to avoid numerical instabilities, the time step $\Delta t$ is chosen according to the recommendations of Tang et al. (1999),

$$\max_i\left(2 k_i x_i\right) \leqslant \Delta t \leqslant \min_i\left(2 k_i(1 - x_i)\right). \tag{8}$$

Equation (8) must be verified for all cells belonging to the entire stream network.

When using lumped stream reaches, Eqs. (2)–(8) have to be adapted as follows: $l_i$ is to be replaced with the reach length, $S_0$
with the average bed slope over the reach, and $Q_{i-1}^n$ with the sum of the outlet discharge(s) of the upstream reach(es) and the
lateral subsurface flow discharge into the stream reach at time $t_n$. In addition, $Q_i^n$ and $h_i^n$ have to be interpreted as the outlet
discharge and mean water depth in the reach at time $t_n$.

Both water routing techniques assume the stream width $w$ to be spatially constant within each reach. Several methods are
available for the computation of $w$, such as for instance a linear function of the total area drained by the stream reach. The
possibility is also offered to set $w$ as a power-law function of the reach outlet discharge, hereby making $w$ time-dependent.
Each of these methods requires the specification of two parameters, which should be set prior to the *StreamFlow* simulation.

### 2.2.2   Stream energy-balance computation

The computation of in-stream temperature assumes a constant cross-sectional profile in each stream reach separately; it is
based on the one-dimensional mass and energy balance equations solved over each stream reach (adapted from Gallice et al.,





2015),

$$\frac{\partial A}{\partial t} + \frac{\partial Q}{\partial x} = q_{subw}, \tag{9}$$

$$\frac{\partial (A T_w)}{\partial t} + \frac{\partial (Q T_w)}{\partial x} = \frac{w\phi}{\rho_w c_{p,w}} + q_{subw} T_{subw} + Q\frac{g}{c_{p,w}} S_0, \tag{10}$$

where $t$ (s) denotes time and $x$ (m) the streamwise distance; $A$ (m$^2$), $Q$ (m$^3$ s$^{-1}$), $T_w$ (K) and $w$ (m) stand for the cross-sectional area, discharge, temperature and width of the stream reach; $\phi$ (W m$^2$) corresponds to the sum of the net heat fluxes at the air–water and water–bed interfaces; $\rho_w$ (kg m$^{-3}$) and $c_{p,w}$ (J kg$^{-1}$ K$^{-1}$) denote the mass density and specific heat capacity of water; $q_{subw}$ (m$^3$ s$^{-1}$ m$^{-1}$) is the lateral subsurface water inflow per unit streamwise distance; and $g$ (m s$^{-2}$) stands for the gravitational acceleration at the Earth's surface. Both $T_{subw}$, the temperature of subsurface water inflow, and $S_0$, the local bed slope, have been defined previously. Equations (9) and (10) are both written in conservative form. Assuming a smooth variation of $A$, $Q$ and $T_w$ along the stream reach, the partial derivatives on the left-hand side of Eq. (10) can be developed using the product rule. By inserting Eq. (9) and re-arranging the terms, one obtains the following expression:

$$\frac{\partial T_w}{\partial x} + v\frac{\partial T_w}{\partial x} = \frac{\phi}{\rho_w c_{p,w} h} + \frac{q_{subw}}{hw}(T_{subw} - T_w) + \frac{gQ}{c_{p,w} hw} S_0, \tag{11}$$

where $v = Q/A$ (m s$^{-1}$) corresponds to the flow velocity and $h = A/w$ (m) to the stream water depth.

In Eqs. (9)–(11), the values of $A$, $Q$, $v$, $h$ and $w$ are provided by the water routing module of *Streamflow* (see Sect. 2.2.1), while $T_{subw}$ is obtained from the subsurface runoff temperature module (see Sect. 2.1.2). The value of $q_{subw}$ is derived from the subsurface runoff discharge $Q_{subw}$ (see Sect. 2.1.1) depending on the stream reach type. In lumped reaches, it is simply computed as $Q_{subw}$ divided by the reach length, whereas it is calculated in each discretized reach cell as the fraction of $Q_{subw}$ proportional to the cell specific drainage area, divided by the cell length.

The net heat flux $\phi$ is computed as in Westhoff et al. (2007) with the following modifications:

1. Incoming short and long wave radiation are directly obtained from meteorological measurements. They are spatially interpolated by *Streamflow* over the stream network using library *MeteoIO* (Bavay and Egger, 2014), taking topographic shading into account. Riparian forest shading is currently not represented in the model, hereby restricting the application of *StreamFlow* to high-altitude catchments. This limitation might be relaxed in the near future through the implementation of an appropriate shade model, taking e.g. advantage of the improvements brought by Gouttevin et al. (2015) to the canopy module of *Snowpack*.

2. The heat flux at the water–bed interface $\phi_b$ (W m$^{-2}$) is computed at any given point along the stream according to Haag and Luce (2008):

$$\phi_b = k_{bed}(T_{bed} - T_w), \tag{12}$$

where $k_{bed}$ (W m$^{-2}$ K$^{-1}$) denotes the bed heat transfer coefficient, which corresponds to the bed heat conductivity multiplied by the distance over which the heat transfer occurs. The value of $k_{bed}$ can be freely specified by the user, but is





fixed here to $52.0 \,\mathrm{W\,m^{-2}\,K^{-1}}$ after Moore et al. (2005) and MacDonald et al. (2014). Stream bed temperature $T_\mathrm{bed}$ (K) is assumed to be equal to soil temperature as modeled by *Alpine3D* at the point of interest, averaged over depth $z_\mathrm{d}$. This depth is the same one as used by the subsurface runoff temperature module (see Sect. 2.1.2) and should be specified prior to running the *Alpine3D* simulation.

3. The latent heat flux $\phi_\mathrm{l}$ $(\mathrm{W\,m^{-2}})$ is approximated using a simplified Penman equation (e.g. Hannah et al., 2004; Haag and Luce, 2008; Magnusson et al., 2012),

$$\phi_\mathrm{l} = -\frac{\rho_\mathrm{a} c_\mathrm{p,a}}{\gamma}\big(a_{vw} v_\mathrm{wind} + b_{vw}\big)\big(e_\mathrm{s}(T_\mathrm{w}) - e(T_\mathrm{a})\big), \tag{13}$$

where $T_\mathrm{a}$ (K), $\rho_\mathrm{a}$ $(\mathrm{kg\,m^{-3}})$ and $c_\mathrm{p,a}$ $(\mathrm{J\,kg^{-1}\,K^{-1}})$ denote the temperature, mass density and specific heat capacity of air, $v_\mathrm{wind}$ $(\mathrm{m\,s^{-1}})$ the wind velocity, $\gamma$ $(\mathrm{Pa\,K^{-1}})$ the psychrometric constant, $e_\mathrm{s}(T_\mathrm{w})$ (Pa) the saturated vapor pressure

measured at stream temperature, and $e(T_\mathrm{a})$ (Pa) the actual vapor pressure measured at air temperature. The values of parameters $a_{vw}$ (–) and $b_{vw}$ $(\mathrm{m\,s^{-1}})$ are chosen after Webb and Zhang (1997), namely $a_{vw} = 2.20 \times 10^{-3}$ and $b_{vw} = 2.08 \times 10^{-3}\,\mathrm{m\,s^{-1}}$, although they can be changed by the user.

4. The sensible heat flux $\phi_\mathrm{h}$ $(\mathrm{W\,m^{-2}})$ is computed based on an approach similar to the one used in Comola et al. (2015), namely as

$$\phi_\mathrm{h} = -\rho_\mathrm{a} c_\mathrm{p,a}\big(a_{vw} v_\mathrm{wind} + b_{vw}\big)\big(T_\mathrm{w} - T_\mathrm{a}\big). \tag{14}$$

This expression for $\phi_\mathrm{h}$ is preferred over the one used in Westhoff et al. (2007), since the latter contains a term $e_\mathrm{s}(T_\mathrm{w}) - e(T_\mathrm{a})$ in the denominator which we observed to be responsible for numerical instabilities when $T_\mathrm{w}$ approaches $T_\mathrm{a}$ (not shown).

In the case of lumped stream reaches, *StreamFlow* uses the first order upwind finite difference approximation of Eqs. (9)–(10)

to estimate stream temperature $T_{\mathrm{w},j}$ in each reach $j$ (see e.g. Westhoff et al., 2007):

$$A_j \frac{\mathrm{d}T_{\mathrm{w},j}}{\mathrm{d}t} = \frac{Q_{\mathrm{in},j}}{L_j}(T_{\mathrm{in},j} - T_{\mathrm{w},j}) + q_{\mathrm{subw},j}(T_{\mathrm{subw},j} - T_\mathrm{w}) + \frac{w_j \phi_j}{\rho_\mathrm{w} c_\mathrm{p,w}} + L_j Q_j \frac{g}{c_\mathrm{p,w}} \overline{S_0}, \tag{15}$$

where $A_j$ $(\mathrm{m^2})$, $Q_j$ $(\mathrm{m^3\,s^{-1}})$, $\overline{S_0}$ (–), $L_j$ (m) and $w_j$ (m) denote the cross-sectional area, outlet discharge, mean bed slope, length and width of reach $j$, and $\phi_j$ $(\mathrm{W\,m^{-2}})$ corresponds to the net heat flux into reach $j$. $Q_{\mathrm{in},j}$ and $T_{\mathrm{in},j}$ stand for the discharge and temperature of water draining into the reach inlet. $Q_{\mathrm{in},j}$ is simply computed as the sum of the outlet discharges of the

upstream reaches, whereas $T_{\mathrm{in},j}$ is approximated as the discharge weighted mean of the outlet temperatures of the upstream reaches. $T_{\mathrm{subw},j}$ and $q_{\mathrm{subw},j}$ denote the temperature and discharge per unit streamwise distance of the subsurface water inflow into reach $j$. Equation (15) is discretized in time using an implicit Euler scheme, whose solution is obtained thanks to the simplified Brent's root finding method proposed by Stage (2013).

In discretized stream reaches, Eq. (11) is solved using a splitting scheme (e.g. LeVeque, 2002). The idea is to decompose the

equation into two simpler ones, where the solution of the first equation serves as initial condition for the second one. Similarly





to Loinaz et al. (2013), we chose here to separate heat advection from the accounting of the heat sources, since standard approaches are available for the numerical resolution of advection in the absence of sources. The resulting splitting scheme is the following (adapted from Loinaz et al., 2013):

$$\frac{\partial T_{\mathrm{w}}}{\partial t} + v\frac{\partial T_{\mathrm{w}}}{\partial x} = 0, \tag{16}$$

$$\frac{\mathrm{d}T_{\mathrm{w}}}{\mathrm{d}t} = \frac{\phi}{\rho_{\mathrm{w}} c_{\mathrm{p,w}} h} + \frac{q_{\mathrm{subw}}}{hw}(T_{\mathrm{subw}} - T_{\mathrm{w}}) + \frac{gQ}{c_{\mathrm{p,w}} hw}S_0. \tag{17}$$

Equation (16) is discretized over each stream reach using an explicit upwind finite volume scheme with second-order precision in space and first-order precision in time (Berger et al., 2005):

$$T_{\mathrm{w},i}^{n+1} = T_{\mathrm{w},i}^{n} - \frac{v_i^n \Delta t}{l_i}\big(T_{\mathrm{w},i+1/2}^{L} - T_{\mathrm{w},i-1/2}^{L}\big). \tag{18}$$

In the above equation, $T_{\mathrm{w},i}^{n}$ (K) and $v_i^n$ ($\mathrm{ms}^{-1}$) denote the stream temperature and flow velocity in reach cell $i$ at time $t_n$, $\Delta t$ corresponds to the time step and $l_i$ is the length of cell $i$. $T_{\mathrm{w},i+1/2}^{L}$ (K) refers to the so-called *left state* at the right boundary of cell $i$, which is computed as:

$$T_{\mathrm{w},i+1/2}^{L} = T_{\mathrm{w},i}^{n} + \frac{1}{2}\psi_i(T_{\mathrm{w},i}^{n} - T_{\mathrm{w},i-1}^{n}), \tag{19}$$

where the factor $\psi_i$ (–), known as a *slope limiter*, is introduced so as to limit numerical dispersion. Many slope limiters have been derived for regular space discretizations (LeVeque, 2002), but very few are available for irregular meshes (Berger et al., 2005; Zeng, 2013). *StreamFlow* implements the slope limiter developed by Zeng (2013),

$$\psi_i = \frac{B(r + r^k)}{1 + Ar^k}, \tag{20}$$

with

$$r = \frac{T_{\mathrm{w},i+1} - T_{\mathrm{w},i}}{T_{\mathrm{w},i} - T_{\mathrm{w},i-1}},$$

$$A = \frac{l_{i-1} + l_i}{l_i + l_{i+1}},$$

$$B = \frac{2l_i}{l_i + l_{i+1}},$$

$$k = \left\lceil \frac{B}{2\min(1, A) - B} \right\rceil.$$

The solution to Eq. (18) is used as initial condition for Eq. (17), which is discretized in time according to an implicit Euler scheme and solved using the root-finding method developed by Stage (2013). A validation of the splitting scheme can be found in Appendix B, where it is compared with analytical solutions to the heat balance equation in two simple test cases.

## 3  Model implementation

In order to allow for the calibration of its parameters, *StreamFlow* was developed as a stand-alone program rather than being seamlessly integrated into *Alpine3D*. This permits a higher flexibility, since *Alpine3D* – whose typical computation time is of





the order of 24 hours when simulating a 1 year period on a standard personal computer – does hereby not need to be newly run each time a new parameter set is tested in *StreamFlow*.

For the sake of consistency, *StreamFlow* is, similarly to *Alpine3D*, implemented in C++ and compiled using CMake. The choice was made to use version C++11 of the C++ language, since it offers new practical features such as anonymous functions or ranged-based for loops as compared to the C++99 standard (Lippman et al., 2012) – regardless of the fact that C++11 is meant to supersede C++99 on the long term. The same coding strategy as detailed in Bavay and Egger (2014) is used here, namely:

– Advantage is taken of the object-oriented nature of C++ to clearly structure the code and make it as modular as possible, so as to facilitate understandability and ease future developments.

– The dependence towards third-party software is avoided as much as possible in order to limit installation issues. The only external utility required by *StreamFlow* is the library *MeteoIO* (Bavay and Egger, 2014), which is used to read input files and interpolate meteorological data in space and time.

– Significant effort is put in documenting the code, both for end-users and future developers. On-line documentation provides indications regarding the installation procedure and the steps to follow in order to launch a simulation (see http://models.slf.ch/p/streamflow/doc/). In addition, technical documentation is directly integrated into the source code using the doxygen tool (van Heesch, 2008).

– Particular attention is paid at keeping the coding style consistent. This task is facilitated by the small size of the development team – mostly one person – and the young age of the project – the creation of *StreamFlow* dates from 2015. The coding style is essentially the same as in *MeteoIO*, with additional conventions regarding the naming of class attributes (see http://models.slf.ch/p/streamflow/page/CodingStyle/).

– When compiling the code, all possible gcc warnings are activated and requested to be passed successfully. The code currently compiles on Windows, Linux and OS X.

– The program is designed so as to be as flexible as possible. In particular, its behavior can be adapted without recompiling the code by modifying the configuration file, which regroups all adjustable parameters. Additionally, the use of library *MeteoIO* for preprocessing allows input data to be provided in a large variety of formats.

– Daily automated tests were set into place using CTest. This ensures that potential errors introduced by code modifications are rapidly identified and corrected, therefore increasing code stability.

The following sections provide some details about the code implementation and the program work flow.



## 3.1 Program main architecture

The program is structured around a main class, *HydrologicalModel*, which is in charge of computing the transport of water and energy within the hillslopes and along the stream network (see Fig. 3a). This class regroups an object of type *Watershed* – representing the catchment – and another one of type *StreamNetwork* – symbolizing the stream network.

Class *Watershed* is nothing but a container storing a collection of *Subwatershed* objects, each one of them representing one of the subcatchments identified by TauDEM. As depicted in Fig. 3a, class *Subwatershed* is subclassed into *LumpedSubwatershedInterface*, which defines the interface to be implemented by lumped subwatersheds – i.e. subwatersheds being treated as single points (see Sect. 2.1.1). Future code developments could include the definition of a second interface inherited from *Subwatershed*, representing the subwatersheds as spatially-distributed entities. Each concrete subclass of *LumpedSubwater-*

*shedInterface* is intended to implement a different approach for calculating the discharge and/or temperature of subsurface runoff (see below).

    Every *Subwatershed* object holds a pointer to its corresponding stream reach, which is represented in the code by class *StreamReach*. The latter is subdivided into two abstract subclasses: *LumpedStreamReachInterface* representing lumped stream reaches, and *DiscretizedStreamReachInterface* symbolizing discretized stream reaches. Each one of these subclasses is further

subclassed into concrete implementations, each implementation corresponding to a specific method for computing stream discharge and/or temperature (see below). All the *StreamReach* objects belonging to the stream network are regrouped into the container class *StreamNetwork*.

    Classes *LumpedSubwatershedInterface*, *LumpedStreamReachInterface* and *DiscretizedStreamReachInterface* are intentionally abstract in order to allow for the implementation of the Decorator pattern. This standard design pattern, illustrated in

Fig. 3b, is aimed at dynamically extending the functionality of a class (Gamma et al., 1994). It is used here to separate the discharge computation from the temperature calculation, which allows each temperature modeling approach to be combined with every discharge computation method. In its commonly accepted definition, the Decorator pattern requires the declaration of a wrapper class – called *ConcreteDecorator* in Fig. 3b – which implements the same interface as the objects to be decorated – called *ConcreteImplementation* in the figure. The presence of abstract class *Decorator* in the pattern (see Fig. 3b) allows

for multiple decorators to be stacked on top of each other, a feature which might be of interest for future developments of *StreamFlow* in case e.g. pollutant transport was to be implemented in the model as an additional decorator. Abstract class *Implementation* is not part of the traditional Decorator pattern, but was introduced in *StreamFlow* in order to implement functionalities which are common to all of its subclasses, hereby reducing duplicate code. In the Decorator pattern, each call to a member function of the wrapper is usually forwarded to the decorated object, with additional operations occurring before

and/or after the forwarded function call. As mentioned above, this pattern is used in *StreamFlow* to separate the computation of discharge from the one of temperature. For example, the concrete subclass of *LumpedSubwatershedInterface*, which implements the linear reservoir model described in Sect. 2.1.1, is only concerned with the modeling of subsurface runoff discharge. The three possible methods detailed in Sect. 2.1.2 for computing subsurface runoff temperature are implemented each in separate decorators of this class. Similarly, some subclasses of *LumpedStreamReachInterface* and *DiscretizedStreamReachInterface*





are in charge of computing stream discharge only; the estimation of stream temperature occurs in the decorators. The interfaces of both decorated and decorator classes – namely classes *Implementation* and *Decorator* in Fig 3b – have been designed in *StreamFlow* so as to be easily extended by a casual developer, therefore facilitating the implementation of future discharge or temperature computation methods.

## 3.2 Input reading

For *StreamFlow* to run properly, *Alpine3D* has to be configured so as to output the grids of the water percolation rate at the bottom of the soil columns. In case stream temperature is to be computed, *StreamFlow* additionally expects grids of soil temperature from *Alpine3D* (see Sect. 2.1.2), on top of the same meteorological measurements as those required by *Alpine3D* as input. These measurements will be interpolated by *MeteoIO* over the stream reaches, taking topographic shading into account in the case of incoming short wave radiation.

Similarly to *MeteoIO*, *StreamFlow* processes its input files in a centralized manner, hereby facilitating the understanding and reuse of the code by casual developers. All required files are parsed by a single class, *InputReader*, which supports various input formats thanks to the integrated use of *MeteoIO* utilities (see Bavay and Egger, 2014). It delegates the actual parsing of the input files to low-end classes, devised to be easily modified or enriched by end users.

## 3.3 Output writing

As a result of its semi-distributed nature, *StreamFlow* is able to output the discharge and temperature of subsurface runoff produced by each subwatershed, as well as the water depth, discharge and temperature in each stream reach. Output files are currently produced in the SMET format (see https://models.slf.ch/docserver/meteoio/SMET_specifications.pdf), for which various utilities – such as parsing and visualizing functions in Matlab and Python – are available in *MeteoIO*. The possibility is offered to the user to generate output files only for certain subwatersheds and/or stream reaches.

As for the parsing of the input files, the writing of the output data is handled by a high-level class, *OutputWriter*, which delegates the actual generation of the output files to low-level classes. As mentioned in the previous section, this architecture both facilitates future developments and eases the understanding of the global code structure.

## 3.4 Calibration module

*StreamFlow* comes with an optimization module used to calibrate the model parameters. It aims to identify the parameter set minimizing the so-called *objective function*. The latter can be freely specified by the user based on the following standard error measures:

– The root mean square error (RMSE)

– The Nash-Sutcliffe efficiency (NSE Nash and Sutcliffe, 1970), also known as the coefficient of determination $R^2$

– The mean absolute error (MAE), corresponding to the average over all time steps of the model error absolute values





– The bias, defined as the mean value of the model errors over all time steps.

Each one of the above four measures can be evaluated either for water depth, discharge or temperature, bringing to a total of 12 the number of different error measures at disposal. *StreamFlow* also supports the case where measurement data is available at more than one point along the stream network. The objective function can be defined as any weighted sum of some (or all)

of the available error measures, hereby making the model calibration entirely flexible. Monte Carlo simulations are currently used for calibrating the model, but other well-known optimization algorithms such as DREAM (Vrugt and Ter Braak, 2011) or GLUE (Beven and Binley, 1992) could be easily integrated into the code.

For the sake of modularity and flexibility, the list of model parameters is not managed centrally in the source code. Instead, each parametrizable class is responsible of defining its own associated parameters. This operation is performed through inher-

itance of a dedicated abstract class, *ParametrizableObject*, which essentially possesses two member functions *getParameters* and *setParameters* for obtaining and modifying the class parameters, respectively. The calibration module can then reconstruct the complete list of model parameters by simply calling method *getParameters* on each object inheriting from *ParametrizableObject*. Based on this list, it can compute new parameter values to be tested, which are transferred back to the individual objects through a call to their method *setParameters*.

In addition to its name, value and units, each model parameter in *StreamFlow* is associated with a range of physically acceptable values and a flag specifying whether it should be calibrated or not. The physically acceptable range is used by the calibration module to restrict the search domain for the best parameter value. The properties of each parameter can be freely set by the user in the program configuration file. In particular, the calibration flag can be individually set to true or false for every parameter, hereby making it possible to calibrate only a given subset of parameters.

**4  Case study**

In view of assessing its accuracy and demonstrating some of its capabilities, *StreamFlow* is tested over a high altitude catchment in Switzerland. Section 4.1 presents the test catchment and the measurement data used to validate the model. The model setup is described in Sect. 4.2 and the simulation results are detailed in Sect. 4.3.

**4.1  Study site and measurement data**

*StreamFlow* is tested over the Dischma catchment, located in the eastern Swiss Alps (see insert in Fig. 4). The gauging station operated by the Swiss Federal Office for the Environment (FOEN) at the location named Davos Kriegesmatte – referred to as *Outlet* in Fig. 4 – is chosen as the catchment outlet. At this point, the watershed has an area of $43.3 \ \mathrm{km}^2$ and is mostly covered with pasture ($36\%$), rock outcrops ($24\%$) and bare soil ($16\%$), with only $2\%$ of glacier cover (Schaefli et al., 2014). Very little riparian vegetation is present along the stream, which ensures that the current absence of riparian shade model in *StreamFlow*

does not have a large influence on the quality of the stream temperature simulation. The watershed elevation ranges from about 1700 m to more than 3100 m above sea level. Its hydrological regime was classified as glacio-nival by Aschwanden and Weingartner (1985), i.e. the stream discharge is low in winter and peaks in June–July due to snow and ice melt, therefore



corresponding to a typical watershed over which *StreamFlow* is meant to be used. More information on the Dischma catchment can be found in e.g. Zappa et al. (2003) and Schaefli et al. (2014).

Water depth, discharge and temperature are continuously monitored by the FOEN at the catchment outlet. In complement to the quality control performed by the FOEN, hourly mean data is also corrected here using the procedure described in Gallice et al. (2015), namely a combination of visual inspection and automatized outlier identification. In addition to the FOEN station, two temporary gauging stations were installed starting on 16 January 2015 at the locations named Am Rin and Dürrboden, indicated as red triangles in Fig. 4. The gauging station at Am Rin was positioned in a small stream coming from a side valley, just above its confluence with the main stream. The station at Dürrboden was deployed in the upper part of the main stream, just below the confluence with the rivulet coming from the glacier. Both stations continuously measured water depth and stream temperature at a rate of one hour. Discharge was manually estimated using the salt dilution technique on a few days during winter and spring, which enabled the derivation of a rating curve to convert the continuous water depth measurements into discharge values (e.g. Weijs et al., 2013). The data from the gauging stations at Am Rin and Dürrboden is corrected using the same protocol as the data provided by the FOEN.

The meteorological data used to run the *Alpine3D* simulation and compute the stream temperature in *StreamFlow* is obtained from two different sources:

(a) The Swiss Federal Office of Meteorology and Climatology, MeteoSwiss, which operates a country-wide network of automatic weather stations. Two of these are in the vicinity of the Dischma catchment: the Davos and Weissfluhjoch stations, whose respective locations are about 5 and 8.5 km on the North-West of the catchment outlet. They are both equipped with heated rain gauges, the one at Davos being unshielded and the one at Weissfluhjoch shielded. These stations provide measurements of air temperature, relative humidity, incoming long and short wave radiation, precipitation, wind direction and snow height every hour.

(b) The Intercantonal Measurement and Information System (IMIS), a network of automated weather stations mostly used for avalanche forecasting in Switzerland (Lehning et al., 1999). Four of these stations are used in the present study, whose distances to the catchment outlet are 0.9, 4.7, 5.9 and 9.5 km. They continuously measure air temperature, relative humidity, outgoing short wave radiation, wind speed and snow depth at a rate of one hour.

All meteorological time series are visually inspected to detect sensor failure. Data flagged as erroneous is removed from the time series.

## 4.2 Model setup

As mentioned previously, *StreamFlow* requires *Alpine3D* to be executed first. In the present case, *Alpine3D* is run over a grid with 100 m resolution and with an internal time step of 15 minutes. The simulated time period extends over three hydrological years, namely from 1$^{st}$ October 2012 to 1$^{st}$ October 2015. All meteorological input data are spatially interpolated using the inverse-distance weighting approach with lapse rate, except for solar radiation and precipitation. Solar radiation is computed based on the measurements at Weissfluhjoch station alone, taking atmospheric attenuation into account for each grid cell sep-



arately. Precipitation is interpolated using the data measured at the Davos station only. It is corrected for undercatch using the approach advocated by the World Meteorological Organization (WMO) for Hellmann gauges (Goodison et al., 1998), before being distributed over each grid cell based on a lapse rate proportional to the measured precipitation intensity at Davos. Another procedure using the data from Weissfluhjoch station in addition to the one from Davos was also tested for interpolating precip-

itation. However, it was rejected since it largely overestimated the total amount of precipitation falling over the catchment, due to the existence of a strong North-South precipitation gradient in the area, making the measurements at Weissfluhjoch station – located further North – less representative of the situation in the Dischma catchment than those at Davos station – located closer to the catchment (Voegeli et al., 2016).

As an additional preliminary step to the *StreamFlow* simulation, the stream network and its corresponding set of subwa-

tersheds are, as described in Sect. 2, extracted from a 25 m resolution DEM provided by the Swiss Federal Office of Topography, SwissTopo (see http://www.swisstopo.admin.ch/internet/swisstopo/en/home/products/height/dhm25.html). Application of the automatic Peuker–Douglas extraction method provided by TauDEM (see http://hydrology.usu.edu/taudem/taudem5/help53/PeukerDouglas.html) results in a subdivision of the catchment into 39 subwatersheds, ranging in size from 0.2 ha to 6.4 km$^2$ (see Fig. 4). It should be mentioned that the difference in resolution between the DEM provided as input to *Alpine3D*

($100 \times 100$ m) and the one used to extract the stream network ($25 \times 25$ m) is seamlessly handled by *StreamFlow*. This allows, as in the present case, for *Alpine3D* to be run over a coarser grid than *StreamFlow*, hereby saving computational power and resources.

*StreamFlow* is configured so as to compute the width $w$ of each stream reach as: $w = a_w A_{\mathrm{reach,tot}} + b_w$, where $A_{\mathrm{reach,tot}}$ (m$^2$) denotes the total area drained by the reach – including its upstream reaches. Parameters $a_w$ (m$^{-1}$) and $b_w$ (m) are determined

approximately based on the width of the main stream estimated at sample locations using aerial photographs of the Dischma catchment. In addition, the values of parameters $\alpha_h$ and $\beta_h$, which are required by the model to estimate water depth when simulating discharge based on the instantaneous advection technique (see Sect. 2.2.1), are derived from the discharge gauging curve provided by the FOEN at the catchment outlet. All model parameters used for the *StreamFlow* simulations presented in the next section are summarized in Table 2, along with their respective calibration ranges when appropriate. For the purpose

of reducing the impact of the initial conditions on the modeled stream variables, a warm-up period of one year is considered. In other words, the model is run over a random year before each simulation, and its state at the end of the warm-up period is used as an initial condition for the actual simulation. The model is calibrated over hydrological year 2013 using Monte-Carlo simulations, and validated over hydrological years 2014 and 2015. Calibration is performed in two steps:

1. All parameters associated with water routing, whether within the hillslopes or along the stream network, are calibrated

by maximizing the Nash-Sutcliffe efficiency of simulated discharge at the catchment outlet. Only the parameters associated with subsurface runoff modeling are actually calibrated in this step (namely $R_{\max}$, $\overline{\tau}_{\mathrm{res,u}}$ and $\overline{\tau}_{\mathrm{res,l}}$), since the only parameter related to water routing within the stream channels (i.e. Manning's coefficient) is fixed to some predefined values (see Sect. 4.3 and Table 2).





2. The parameters calibrated in step 1 are kept fixed to their respective best values, while the parameters related to stream temperature modeling are calibrated by maximizing the NSE of simulated temperature at the catchment outlet. This step is repeated for each method used to compute the temperature of subsurface runoff (see Sect. 2.1.2). The parameters associated with the water heat balance in the stream network are fixed to specific values based on physical considerations (see Table 2).

In order to better assess the accuracy of *StreamFlow*, the approach advocated by Schaefli and Gupta (2007) is followed here. The error measures associated with *StreamFlow* are compared to those of a simplistic benchmark model, so as to verify whether *StreamFlow* allows for more robust predictions than those that could be made based on a basic procedure. Two benchmark models are actually considered here, one for discharge and one for temperature. Both are constructed by averaging, for each hour of each day of the year, the values of discharge and temperature measured at the catchment outlet on those particular hour and day over a period of 10 years (2005–2014). Stated otherwise, the two models correspond to the measured yearly curves of mean hourly discharge and temperature at the catchment outlet, averaged over ten years.

### 4.3 Model evaluation

#### 4.3.1 Results of the *Alpine3D* simulation

The *Alpine3D* simulation is observed to rather accurately capture the time evolution of the snow pack. As an example, Fig. 5 depicts the simulated snow depth in comparison with the measured one at the Stillberg meteorological station, which is located at an altitude of 2085 m above sea level on the Western slope of the catchment. It can be noticed that the onset of snow accumulation and the timing of the melting period are satisfyingly reproduced, in addition to the fact that the snow depth appears to be overall well simulated. A more quantitative assessment of the accuracy of the *Alpine3D* simulation is obtained by considering the global volume of water transiting through the watershed each year. Thus, the measured cumulated volume of water $V_{\text{out,meas}}$ flowing through the catchment outlet each year is compared to the simulated cumulated volume of water $V_{\text{in,simu}}$ percolating at the bottom of all the soil columns belonging to the watershed over the same year. As can be observed in Table 3, the relative difference between $V_{\text{out,meas}}$ and $V_{\text{out,simu}}$ remains within the range $\pm 8\%$ for all three hydrological years.

#### 4.3.2 *StreamFlow* simulations of discharge and water depth

As mentioned in the previous section, *StreamFlow* parameters related to discharge computation are calibrated against measured discharge at the catchment outlet. To this end, 10 000 Monte-Carlo simulations are run, with *StreamFlow* configured so as to use a time step of 1 hour and advect water in the stream channels based on the instantaneous routing scheme (see Sect. 2.2.1). Figure 6 presents a comparison of the simulated and measured hourly mean discharges over the three considered hydrological years. The uncertainty range of the simulated curve is defined by all parameter sets associated with a NSE larger than 0.85 during the calibration period, which amounts to a total of 300 curves. As observed in panel (a), the simulation corresponding to the highest NSE value matches globally well with the observations, except for a few discharge peaks which are not well captured in 2013 and 2015. The simulation uncertainty range appears to be relatively narrow on an annual scale. When looking





at a finer scale, it can be observed that the daily fluctuations of discharge are relatively well captured by the model, as for example shown in panel (b) for the period 29 May to 8 June 2015. On the other hand, the absence of a fast runoff component in *StreamFlow* prevents the model to correctly capture short-lived discharge peaks. As displayed in panel (c), the modeled recession in these cases is much too slow compared to the observed one.

5    Table 4 presents quantitative error measures of discharge modeled over the validation period at the three gauging points located in the Dischma catchment (see Fig. 4), for the same *StreamFlow* configuration as in Fig. 6. The accuracy of the benchmark model at the catchment outlet is also indicated in the table for comparison. It should be mentioned that the benchmark model could not be evaluated at the two intermediate stations since the measurement time series at these points extend over less than a year (see Sect. 4.1). As observed in the table, the discharge NSE value associated with the best *StreamFlow* simulation is larger than $0.80$ at all three points, as opposed to the NSE value of the benchmark model not exceeding $0.74$. On the other hand, the values of NSE-log – defined as NSE computed with the logarithm of the discharge values – are quite comparable between both models. This is not particularly surprising in view of the strong seasonality of the baseflow component of discharge, particularly during the winter season. The NSE-log value at point Am Rin is rather low, but should be considered with caution since the discharge gauging curve at this point is rather uncertain (not shown). The bias of *StreamFlow* is observed to be positive at all three gauging points, which certainly results from the slight overestimation of the rate of water percolating at the bottom of the soil columns in the *Alpine3D* simulation (see above). Overall, the performance of *StreamFlow* regarding discharge computation based on the instantaneous water routing scheme can be considered as satisfying. Its accuracy is comparable to the one of other existing hydrological models applied over high Alpine catchments (e.g. MacDonald et al., 2014; Schaefli et al., 2014).

20    Regarding the calibration parameters, it appears that the values of $R_\mathrm{max}$ and $\overline{\tau}_\mathrm{res,u}$ are rather well identified (see Fig. 7). Indeed, their respective distributions based on the best 300 parameter sets are contained within a rather narrow interval, clearly separated from the bounds of the respective calibration ranges. Within this interval however, the two parameters are strongly correlated with one another, as pictured in panel (c) of Fig. 7. This points at the equifinality of the parameter sets (Bárdossy, 2007), since an increase in $\overline{\tau}_\mathrm{res,u}$ conjugated with a decrease in $R_\mathrm{max}$ maintains the model accuracy almost constant. As opposed to $\overline{\tau}_\mathrm{res,u}$ and $R_\mathrm{max}$, parameter $\overline{\tau}_\mathrm{res,l}$ is associated with a broad distribution, sticking to the upper boundary of the calibration interval (see panel (a) of Fig. 7). As such, *StreamFlow* appears to be relatively insensitive to the value of $\overline{\tau}_\mathrm{res,l}$, as further emphasized by the low correlation between $\overline{\tau}_\mathrm{res,l}$ and the other two parameters (Bárdossy, 2007).

In order to evaluate the influence of the channel water routing scheme on the modeled discharge, *StreamFlow* was run with the following configurations in complement to the instantaneous routing technique evaluated above: (a) the Muskingum-Cunge approach with lumped stream reaches and Manning's coefficient $n_\mathrm{m}$ set to $0.04$, (b) same as (a) but with $n_\mathrm{m} = 0.10$, and (c)–(d) same as (a)–(b) but with discretized stream reaches. The chosen values for Manning's coefficient correspond to the lower and upper boundaries of the uncertainty interval estimated in the Dischma catchment based on the work of Phillips and Tadayon (2006). The results indicate that the modeled hourly mean discharge curves in all cases (a) to (d) almost identically correspond to the one depicted in Fig. 6, up to a maximum RMSE of $0.03 \ \mathrm{m^3\,s^{-1}}$ between all curves over the entire simulated period (not shown). Similarly, the error measures reported in Table 4 are also valid in cases (a) to (d). The routing technique therefore



appears to have only a very limited impact on the simulated discharge in the Dischma catchment, which is easily explained by the small size of the watershed (Schaefli et al., 2014). Indeed, the average streamwise distance between the stream cells and the catchment outlet is about 6.6 km, which – assuming a flow velocity of $1 \, \mathrm{m \, s^{-1}}$ – corresponds to a mean travel time of about 2 hours down to the catchment outlet. This also explains the observed low sensitivity of *StreamFlow* to the value of Manning's

coefficient in the present case. As expected, the above results suggest that, in small to medium-sized catchments, the use of spatially discretized stream reaches to simulate discharge is not associated with any marked improvement with respect to the lumped approach.

Albeit discharge is simulated unequivocally by all water routing techniques, water depth and flow velocity are not. As pictured in Fig. 8 for hydrological year 2014, differences between the simulated water depth curves are quite large, with for

example a RMSE of 44.5 cm between the curve associated with the instantaneous routing technique and the one corresponding to the Muskingum-Cunge approach with $n_\mathrm{m} = 0.04$. The instantaneous water routing technique predicts here a higher water depth as compared to the Muskingum-Cunge approach, reflecting the values of the gauging curve coefficients $\alpha_h$ and $\beta_h$ (see Sect. 2.2.1 and Table 2). In addition, the predictions based on the Muskingum-Cunge technique depend on the value of Manning's coefficient, as expected from Eq. (7): the higher $n_\mathrm{m}$, the higher the simulated water depth. However, as for the case

of discharge, the water depth estimations do not appear to benefit from the use of discretized stream reaches as opposed to lumped ones, since both corresponding curves almost overlap for a fixed $n_\mathrm{m}$ (see legend of Fig. 8). It should be mentioned that comparison with the measured water depth is hazardous since the modeled river width at the outlet gauging station was not verified to correspond to the observed one. The measured curve is therefore only shown here as an indication. The fact that it diverges from the curve associated with the instantaneous advection approach during winter is due to the fact that the discharge

gauging curve of FOEN is linear for small water depth values, and adopts the form of a power function as in *StreamFlow* only for larger values of $h$. Given that simulated discharge is almost the same for all water routing techniques, the differences in simulated water depth result in large differences in the simulated flow velocity as well (not shown).

### 4.3.3 *StreamFlow* simulations of stream temperature

Turning now to the stream temperature simulations, we first determine an appropriate value for the soil temperature averaging

depth $z_\mathrm{d}$ (see Sect. 2.1.2 and Eq. (12)). Five different possibilities are considered here: 0.15, 0.30, 0.60, 1.20 and 2.40 m. Using *StreamFlow* configured so as to approximate the temperature of subsurface runoff as the depth-averaged soil temperature (Sect. 2.1.2), we find that the choice $z_\mathrm{d} = 2.40$ m leads to the best results in terms of temperature-based NSE (not shown). This rather large value may be due to the relatively low resolution of the vertical soil temperature profile computed by *Alpine3D*, which was configured here to use a coarse vertical discretization of the soil columns in order to spare computational power. The

value $z_\mathrm{d} = 2.40$ m is nevertheless used in the remaining of this study, since emphasis is on demonstrating the model capabilities rather than performing particularly accurate simulations.

Figure 9 displays stream temperature as simulated by *StreamFlow* over the hydrological years 2013–2015, with channel water being advected based on the instantaneous routing scheme and subwatershed outflow temperature being approximated as the depth-averaged soil temperature (same configuration as above). As evident from panel (a), stream temperature is generally



underestimated by the model on a daily time scale, particularly during the snow melt season in spring. This may be due to the simulated soil temperature being too low, since its value averaged down to $2.40$ m typically remains around $0\,^\circ$C until mid-June (not shown). Soil temperature is then modeled by *Alpine3D* to rapidly increase past the snowmelt season, which might explain the better agreement between the simulated and measured stream temperature curves during summer. Panels (b) and (c) present

a zoom on two selected periods during winter and summer, respectively. *StreamFlow* is observed to be capable of simulating the diurnal cycle of stream temperature, albeit its magnitude is in general too low. It should be specified that temperature is cut off at $0\,^\circ$C by the model in winter in order to avoid unphysical values. The underestimation of the magnitude of the diurnal cycle may originate from an overestimation of water depth or, equivalently, from an underestimation of the stream width. This hypothesis can unfortunately hardly be tested, since water depth and stream width are difficult to quantify in small mountainous

streams with irregular, boulder-covered beds. We verified whether the latent and sensible heat fluxes are not underestimated by *StreamFlow*. To this end, we increased the values of coefficients $a_{vw}$ and $b_{vw}$ by $50\%$ (see Eq. (13) and Table 2), however this did not result in a marked improvement of the simulated diurnal cycle (not shown). The effect of the heat exchange with the stream bed was also tested by reducing the value of $k_{\text{bed}}$ by $50\%$ (see Eq. 12 and Table 2), but this had almost no impact on the simulated temperature curve either (not shown). The underestimation of the diurnal cycle therefore appears to mostly

originate from the approach selected for the modeling of subsurface runoff temperature, as discussed into more detail below. From the inspection of all three panels in Fig. 9, it can be stated that modeled temperature is not particularly affected by the uncertainty in the values of hydrological parameters $R_{\text{max}}$, $\overline{\tau}_{\text{res,u}}$ and $\overline{\tau}_{\text{res,l}}$. As a matter of fact, the uncertainty range of the simulated temperature curve remains globally narrow, except around midday where it reaches a value up to $1\,^\circ$C on some days (see panel (b)). This limited sensitivity of modeled temperature with respect to simulated discharge (and water depth) further

hints at the probable role of subsurface runoff temperature on the underestimation of the temperature diurnal cycle.

The values of the error measures associated with Fig. 9 are summarized in Table 5. The NSE value of the hourly mean temperature curve ($0.78$) is much lower than the one of the benchmark model ($0.87$), which denotes a strong improvement potential. This has to be put into perspective with the fact that the Dischma river is rather small and heavily turbulent, and therefore more challenging to model as compared to larger, low altitude rivers. In addition, the NSE value is comparable

to the one reported by MacDonald et al. (2014) over a mountainous watershed of similar size and altitudinal range as the Dischma catchment. The RMSE equals $1.45\,^\circ$C, which is not very far from the RMSE of the benchmark model ($1.14\,^\circ$C). On the other hand, the bias is rather large ($-0.88\,^\circ$C), as already noted from the observation of Fig 9 above. Regarding the model performance at the two intermediate gauging points, the values of the error measures at Dürrboden are found to be essentially the same as at the outlet point, except for the positive bias (see Table 5). Concerning Am Rin, the apparent better values for

RMSE, NSE and bias have to be weighted against the short time period over which they are evaluated (17 January 2015 to 17 July 2015).

As already discussed above, the simulated stream temperature is not particularly sensitive to the modeled discharge. This fact is confirmed by the values of the error measures reported in Table 5 for four temperature simulations, each one based on a different water routing scheme – namely the instantaneous advection technique or the Muskingum-Cunge approach, combined with either a lumped or discretized modeling of the stream reaches. In the simulations based on the Muskingum-



Cunge approach, Manning's coefficient is fixed to $0.07$, which corresponds to the middle of the above-defined range of plausible values in the case of the Dischma. It appears that all four simulations are associated with a similar accuracy in terms of stream temperature modeling, as indicated by the narrow range of NSE ($0.77$–$0.80$) and RMSE ($1.40$–$1.49\,^\circ$C) values. Contrary to the discharge simulations, the discretized representation of the stream reaches enables here a slight improvement of the results as compared to the lumped approach, mainly due to a better resolution of the diurnal cycle (not shown).

In a recent study, Leach and Moore (2015) reviewed the approaches implemented in some of the most popular stream temperature models for approximating the temperature of subsurface runoff. Based on a comparison with data collected in a small Canadian watershed, they concluded that none of them performed well, except for the method implemented in the HSPF model approaching the observations relatively closely. More interestingly, the authors pointed at large discrepancies between the predictions of the various models. As a further step, we propose here to investigate the effect of modeled subsurface runoff temperature on the simulated stream temperature at the catchment outlet. To this end, three *StreamFlow* simulations are run with the same configuration as above – namely lumped reaches and the instantaneous routing scheme – except that the temperature of subsurface runoff is computed each time based on a different method out of the three available ones (see Sect. 2.1.2). It should be mentioned that, in virtue of the modular structure of *StreamFlow*, changing from one method to the next simply requires one line to be modified in the configuration file. The simulation results are displayed in Fig. 10, and the corresponding error measures can be found in Table 5. It can be observed that the approach used to compute the temperature of subsurface runoff has a strong influence on the accuracy of the modeled stream temperature. The method originally implemented in *StreamFlow* appears to perform worse (NSE of $0.56$, RMSE of $2.06\,^\circ$C), followed by the HSPF approach (NSE of $0.70$, RMSE of $1.69\,^\circ$C). The method based on the depth-averaged soil temperature is associated with the best performance measures (see above). Overall, the three methods seem to determine the temperature of in-stream water to a large extent, leading to variations of more than $4\,^\circ$C between the different curves (see Fig. 10). These observations point at the strong need for additional field investigations of the dynamics of subsurface runoff temperature, as already mentioned by Leach and Moore (2015).

## 5 Conclusions

Combined modeling of hydrological and thermodynamic processes offers promising perspectives for the prediction of stream temperature at the catchment scale. The present study describes a new coupled hydro-thermal model, named *StreamFlow*, which is currently intended to be used in high Alpine environments. Designed as an independent extension to the spatially-distributed snow model *Alpine3D*, it has been written entirely anew compared to its initial version. The resulting code has a clear and modular structure which takes advantage of some of the latest available object-oriented features. Several of the hydrological processes represented in the model can be simulated using various alternatives. For example, the advection of water in the stream channels can be computed using either the Muskingum-Cunge technique or an instantaneous routing approach. This modularity enables the model to be adapted to the specific needs of each user, but also provides a rapid means to estimate the uncertainty of the simulation results by comparing the predictions of the various modeling alternatives.





Based on an evaluation over a high Alpine catchment, the model accuracy is shown to be satisfying, with Nash-Sutcliffe efficiencies for the hourly mean discharge and hourly mean temperature being equal to $0.82$ and $0.78$, respectively. The various water routing techniques available in *StreamFlow* do not appear to have a marked effect on the quality of the simulations. On the other hand, it was observed that the approach used to compute the temperature of subsurface runoff strongly impacts the simulated stream temperature at the catchment outlet. This effect has not been reported in any previous study and points at the need for more intensive field investigations of subsurface runoff temperature.

Several improvements can be brought to the actual state of the model. The representation of riparian shading would allow *StreamFlow* to be applied in lower-altitude, vegetated watersheds. However, similarly to the case of subsurface runoff temperature, the shading by riparian vegetation is a complex phenomenon which is difficult to simulate and requires further research (Moore et al., 2005). The modeling of the ice and snow sheet forming over the stream in winter could also be included in *StreamFlow*, using for example an approach similar to the one introduced by van Beek et al. (2012). Finally, additional modeling alternatives could be implemented for various components of *StreamFlow*, such as the approach developed by Leach and Moore (2015) for the estimation of subsurface runoff temperature, or the full St-Venant equations for the routing of water in the stream channels.

In the near future, we plan to use *StreamFlow* in order to evaluate the effects of climate change on the hydrological functioning of high alpine watersheds. In particular, advantage will be taken of the coupled hydro-thermal nature of the model in order to investigate the impact of the future discharge modifications on stream temperature.

### Code availability

The source code of *StreamFlow* is available under the GNU Lesser General Public License v3.0 (LGPL v3) at http://models. slf.ch/p/streamflow/ upon creation of a free account. Installation instructions can be found at http://models.slf.ch/p/streamflow/ page/Installing-StreamFlow/, and the detailed procedure to launch a *StreamFlow* simulation at http://models.slf.ch/p/streamflow/ page/Running-StreamFlow/.

### Appendix A: Formulation of the subwatershed linear reservoir model

This section briefly describes the approaches which were already present in the original version of the code for computing the discharge and temperature of the subsurface flow generated by each subwatershed.

### A1 Subwatershed outflow discharge computation

As illustrated in Fig. 11a, the original model developed by Comola et al. (2015) approximates each subwatershed as the vertical superposition of two linear reservoirs, where the upper one simulates the fast response to rainfall events and the lower one the slow response. Water percolating at the bottom of the subwatershed soil columns fills the lower reservoir up to a maximum flow rate $R_{\max}$ ($\mathrm{m\,s^{-1}}$), the excess water draining into the upper reservoir. This translates into the following equations for the





water levels $S_{\mathrm{res,u}}$ (m) and $S_{\mathrm{res,l}}$ (m) in the upper and lower reservoirs, respectively:

$$\frac{\mathrm{d}S_{\mathrm{res,u}}}{\mathrm{d}t} = I_{\mathrm{res,u}} - \frac{Q_{\mathrm{res,u}}}{A_{\mathrm{subw}}}, \tag{A1}$$

$$\frac{\mathrm{d}S_{\mathrm{res,l}}}{\mathrm{d}t} = I_{\mathrm{res,l}} - \frac{Q_{\mathrm{res,l}}}{A_{\mathrm{subw}}}, \tag{A2}$$

where the water inflow rates $I_{\mathrm{res,u}}$ $(\mathrm{m\,s^{-1}})$ and $I_{\mathrm{res,l}}$ $(\mathrm{m\,s^{-1}})$ into the upper and lower reservoirs are expressed as $I_{\mathrm{res,u}} = I - I_{\mathrm{res,l}}$

and $I_{\mathrm{res,l}} = \min\left(I, R_{\max}\right)$, with $I$ $(\mathrm{m\,s^{-1}})$ denoting the total flow rate of water percolating at the bottom of the subwatershed soil columns and $A_{\mathrm{subw}}$ $(\mathrm{m^2})$ the subwatershed surface area. $Q_{\mathrm{res,u}}$ $(\mathrm{m^3\,s^{-1}})$ and $Q_{\mathrm{res,d}}$ $(\mathrm{m^3\,s^{-1}})$ correspond to the discharge at the outlet of the upper and lower reservoirs, which are linearly related to the reservoir water levels,

$$Q_{\mathrm{res,u}} = A_{\mathrm{subw}} \frac{S_{\mathrm{res,u}}}{\tau_{\mathrm{res,u}}}, \tag{A3}$$

$$Q_{\mathrm{res,l}} = A_{\mathrm{subw}} \frac{S_{\mathrm{res,l}}}{\tau_{\mathrm{res,l}}}. \tag{A4}$$

The characteristic residence times $\tau_{\mathrm{res,u}}$ (s) and $\tau_{\mathrm{res,l}}$ (s) are expressed as power functions of the subwatershed area:

$$\tau_{\mathrm{res,u}} = \overline{\tau}_{\mathrm{res,u}} \left(\frac{A_{\mathrm{subw}}}{A_{\mathrm{tot}}}\right)^{\frac{1}{3}}, \tag{A5}$$

$$\tau_{\mathrm{res,l}} = \overline{\tau}_{\mathrm{res,l}} \left(\frac{A_{\mathrm{subw}}}{A_{\mathrm{tot}}}\right)^{\frac{1}{3}}, \tag{A6}$$

where $\overline{\tau}_{\mathrm{res,u}}$ (s) and $\overline{\tau}_{\mathrm{res,l}}$ (s) are two user-specified parameters and $A_{\mathrm{tot}}$ $(\mathrm{m^2})$ denotes the area of the entire parent watershed. The total discharge $Q_{\mathrm{subw}}$ $(\mathrm{m^3\,s^{-1}})$ flowing from the subwatershed into the stream is then computed as $Q_{\mathrm{subw}} = Q_{\mathrm{res,u}} + Q_{\mathrm{res,l}}$.

The subwatershed behavior can be adjusted by modifying the values of parameters $R_{\max}$, $\overline{\tau}_{\mathrm{res,u}}$ and $\overline{\tau}_{\mathrm{res,l}}$.

## A2 Subwatershed outflow temperature computation

The method developed by Comola et al. (2015) for the computation of the subwatershed outflow temperature $T_{\mathrm{subw}}$ (K) is depicted in Fig. 11b. Temperatures $T_{\mathrm{res,u}}$ (K) and $T_{\mathrm{res,l}}$ (K) of water stored in the upper and lower reservoirs are computed as:

$$\frac{\mathrm{d}T_{\mathrm{res,u}}}{\mathrm{d}t} = \frac{I_{\mathrm{res,u}}}{S_{\mathrm{res,u}}}(T_{\mathrm{soil}} - T_{\mathrm{res,u}}) + \frac{T_{\mathrm{soil}} - T_{\mathrm{res,u}}}{k_{\mathrm{soil}}}, \tag{A7}$$

$$\frac{\mathrm{d}T_{\mathrm{res,l}}}{\mathrm{d}t} = \frac{I_{\mathrm{res,l}}}{S_{\mathrm{res,l}}}(T_{\mathrm{soil}} - T_{\mathrm{res,l}}) + \frac{\overline{T}_{\mathrm{soil}} - T_{\mathrm{res,u}}}{k_{\mathrm{soil}}}, \tag{A8}$$

where $k_{\mathrm{soil}}$ (s) is a calibration parameter corresponding to the characteristic time of thermal diffusion and $T_{\mathrm{soil}}$ (K) refers to soil temperature at the bottom of the subwatershed soil columns as modeled by *Alpine3D*. $\overline{T}_{\mathrm{soil}}$ denotes the annual average of $T_{\mathrm{soil}}$, which is used as a proxy for the temperature of deep soil. The first term in the right hand-side of the above two expressions accounts for the heat flux associated with the inflow of water into the reservoirs. The second term corresponds to

the diffusive heat exchange between water and the surrounding soil particles. These expressions were derived by assuming that the temperature of water percolating at the bottom of the soil columns is equal to the local soil temperature. They are solved





using a second-order Crank Nicholson scheme, and their solution is used to compute $T_{\text{subw}}$ (K) as the weighted average of $T_{\text{res,u}}$ and $T_{\text{res,l}}$:

$$T_{\text{subw}} = \frac{Q_{\text{res,u}} T_{\text{res,u}} + Q_{\text{res,l}} T_{\text{res,l}}}{Q_{\text{res,u}} + Q_{\text{res,l}}}. \tag{A9}$$

**Appendix B: Validation of the splitting scheme used to solve the heat balance equation**

5   The splitting scheme described in Sect. 2.2.2 for numerically solving Eq. (11) is validated here by comparing its predictions against analytical solutions. The derivation of the analytical solutions is presented first, followed by the assessment the numerical scheme precision.

**B1   Analytical solutions to the heat balance equation**

Eq. (11) can be written in a more compact form:

$$\frac{\partial T_{\text{w}}}{\partial x} + v \frac{\partial T_{\text{w}}}{\partial x} = \frac{1}{\tau} T_{\text{w}} + \sigma, \tag{B1}$$

with

$$\tau = -\frac{hw}{q_{\text{subw}}},$$

$$\sigma = \frac{\phi}{\rho_{\text{w}} c_{\text{p,w}} h} + \frac{q_{\text{subw}}}{hw} T_{\text{subw}} + \frac{gQ}{c_{\text{p,w}} hw} S_0.$$

Similarly to (e.g. Lowney, 2000), Eq. (B1) above is simplified by assuming $\tau$ to be constant and $\sigma$ to be a sole function of 15   time. The length of the spatial domain over which the equation is to be solved is denoted as $L$. It is assumed that $v > 0$ for all $x \in [0, L]$, so that a boundary condition must be specified at $x = 0$. A Dirichlet boundary condition is considered here,

$$T_{\text{w}}(0, t) = T_{\text{in}}(t) \qquad \text{for all } t \geqslant 0, \tag{B2}$$

where $T_{\text{in}}(t)$ is a prescribed function of time. Since the spatial domain is finite, the analytical solution to Eq. (B1), subject to boundary condition Eq. (B2), will consist of a transient phase followed by a permanent regime. During the transient phase, 20   the initial temperature distribution $T_{\text{w,ini}}(x, t)$ is advected towards the right end of the spatial domain, while the boundary condition $T_{\text{in}}$ dictates the value of temperature entering the domain through its left-hand end. After the last remnant of the initial temperature distribution has exited the spatial domain, the solution reaches its permanent regime, which is the same regardless of the initial distribution. Only the permanent regime is considered here, so that no initial condition needs to be specified.

25   The analytical solution to Eqs. (B1)–(B2), under the conditions $\tau = \text{cst}$ and $\sigma = \sigma(t)$, is obtained by the method of characteristics (e.g. LeVeque, 2002). The two independent variables $x$ and $t$ are parametrized as a function of a path variable $s$. Using the definition $\theta(s) = T\big(x(s), t(s)\big)$, we observe that

$$\frac{\mathrm{d}\theta}{\mathrm{d}s} = \frac{\partial T_{\text{w}}}{\partial t} \frac{\mathrm{d}t}{\mathrm{d}s} + \frac{\partial T_{\text{w}}}{\partial x} \frac{\mathrm{d}x}{\mathrm{d}s},$$





so that Eq. (11) can be re-written as

$$\frac{\mathrm{d}\theta}{\mathrm{d}s} = \frac{1}{\tau}\theta + \sigma, \tag{B3}$$

if the parametrizations of $x$ and $t$ are chosen such that:

$$\frac{\mathrm{d}t}{\mathrm{d}s} = 1, \tag{B4}$$

$$\frac{\mathrm{d}x}{\mathrm{d}s} = v. \tag{B5}$$

Equation (B3) is an ordinary differential equation in which $\sigma$ should be understood as a function of $s$, i.e. $\sigma(s) = \sigma\big(t(s)\big)$. Its solution can be easily found and is given by:

$$\theta(s) = \int_{s_0}^{s} \left(\sigma(s') + \frac{\theta(s_0)}{\tau}\right) \exp\left(\frac{s - s'}{\tau}\right) \mathrm{d}s' + \theta(s_0), \tag{B6}$$

where $s_0$ denotes the lower integration bound, which needs to be specified. Equation (B4) is trivially solved through integration between $s_0$ and $s$,

$$t(s) = s + s_0 - t_0,$$

where $t_0 = t(s_0)$. The above expression for $t$ implies that $s$ is equivalent to time (i.e. $s \equiv t$), so that $x$ can be interpreted as the position of a particle moving with instantaneous velocity $v$ as per Eq. (B5). In the permanent regime, each "particle" enters the spatial domain through its left-hand side boundary. As a consequence, $s_0$ – or, equivalently, $t_0$ – needs to be chosen such that $x(s_0) = 0$ in the present case. This further implies that:

$$\theta(s_0) = T_{\mathrm{w}}\big(x(s_0), t(s_0)\big) = T_{\mathrm{w}}(0, t_0) = T_{\mathrm{in}}(t_0), \tag{B7}$$

where Eq. B2 has been used in the last step. Inserting the above expression in Eq. (B6) and replacing $\theta(s)$ with $T_{\mathrm{w}}(x,t)$ and $s$ with $t$, one finally obtains:

$$T_{\mathrm{w}}(x,t) = \int_{t_0}^{t} \left(\sigma(t') + \frac{T_{\mathrm{in}}(t_0)}{\tau}\right) \exp\left(\frac{t - t'}{\tau}\right) \mathrm{d}s' + T_{\mathrm{in}}(t_0). \tag{B8}$$

Closed-form expressions of the above equation can be found by choosing simple formulations for $\sigma$ and $v$. Two cases are considered here:

**Test case 1:** Constant velocity and sinusoidal expression for $\sigma$,

$$v(x,t) = \mathrm{cst}, \qquad\qquad \text{for all } x \in [0, L],\, t \geqslant 0, \tag{B9}$$

$$\sigma(t) = a_\sigma \sin(\omega t) + b_\sigma, \qquad\qquad \text{for all } t \geqslant 0, \tag{B10}$$

with $\omega$ $(\mathrm{s}^{-1})$, $a_\sigma$ $(\mathrm{K}\,\mathrm{s}^{-1})$ and $b_\sigma$ $(\mathrm{K}\,\mathrm{s}^{-1})$ constant. This test aims at assessing the ability of the splitting scheme to correctly account for time varying heat sources.





**Test case 2:** velocity varying linearly in space and no $\sigma$-term,

$$v(x,t) = a_v x + b_v, \qquad \text{for all } x \in [0, L], \, t \geqslant 0, \tag{B11}$$

$$\sigma(t) = 0, \qquad \text{for all } t \geqslant 0, \tag{B12}$$

where $a_v$ (s$^{-1}$) and $b_v$ (m s$^{-1}$) are constant and chosen such that $v > 0$ for all $x \in [0, L]$. This test intends to validate the robustness of the splitting scheme in the case of non-uniform flow velocity profiles.

In both cases, the expression of $T_{\text{in}}$ is chosen similarly to the one of (e.g. Lowney, 2000), who aimed at reproducing natural diurnal variations of stream temperature,

$$T_{\text{in}}(t) = a_{\text{in}} \sin(\omega t) + b_{\text{in}}, \tag{B13}$$

where $a_{\text{in}}$ (K) and $b_{\text{in}}$ (K) are constant, and $\omega$ is the same as in Eq. (B10).

### B1.1 Analytical solution of test case 1

In test case 1, the solution to Eq. (B5) under the constraint $x(s_0) = 0$ is straightforward due to $v$ being constant,

$$x(s) = v(s - s_0).$$

Replacing $s$ with $t$ and solving for $t_0$, one obtains:

$$t_0 = t - \frac{x}{v}.$$

After inserting this expression in Eq. B8, replacing $\sigma$ with its sinusoidal formulation and performing the integration, one gets the closed-form expression of the solution to Eq. (11) in the permanent regime (i.e. for $t > L/v$),

$$
\begin{aligned}
T_{\text{w}}(x,t) = {} & T_{\text{in}}\left(t - \frac{x}{v}\right) \exp\left(\frac{x}{\tau v}\right) + b_\sigma \tau \left[\exp\left(\frac{x}{\tau v}\right) - 1\right] \\
& - \frac{a_\sigma \tau}{1 + (\tau \omega)^2} \left(\sin(\omega t) + \tau \omega \cos(\omega t)\right) \\
& + \frac{a_\sigma \tau}{1 + (\tau \omega)^2} \left(\sin\left[\omega\left(t - \frac{x}{v}\right)\right] + \tau \omega \cos\left[\omega\left(t - \frac{x}{v}\right)\right]\right) \exp\left(\frac{x}{\tau v}\right),
\end{aligned}
\tag{B14}
$$

with $T_{\text{in}}$ as defined in Eq. (B13). It should be mentioned that the above expression is actually valid for any formulation of $T_{\text{in}}$, not just Eq. (B13).

### B1.2 Analytical solution of test case 2

In case $v$ is expressed as in Eq. (B11), the solution to Eq. (B5) satisfying $x(s_0) = 0$ becomes:

$$x(s) = \frac{b_v}{a_v}\left(\exp[a_v(s - s_0)] - 1\right).$$





The expression for $t_0$ is obtained by replacing $s$ with $t$ in the above equation:

$$t_0 = t - \frac{1}{a_v} \ln\left(\frac{a_v}{b_v} x + 1\right).$$

The analytical solution of test case 2 is obtained by inserting the above expression for $t_0$ in Eq. (B8), imposing $\sigma = 0$ and performing the integration:

$$T_{\mathrm{w}}(x,t) = T_{\mathrm{in}}\left(t - \frac{1}{a_v} \ln\left(\frac{a_v}{b_v} x + 1\right)\right)\left(\frac{a_v}{b_v} x + 1\right)^{1/(a_v \tau)}. \tag{B15}$$

The above solution describes the permanent regime, i.e. it is valid for all $t \geqslant \ln(a_v L/b_v + 1)/a_v$. As opposed to the solution of test case 1, which has already been reported by Lowney (2000), the present one has – to the best of our knowledge – not been presented in any publication to date.

### B2 Validation of the numerical splitting scheme

The splitting scheme is validated over a spatial domain of $L = 12.8$ km, for a simulated time period of 8 hours. Table 6 contains the values of the parameters considered in test cases 1 and 2.

Figure 12 pictures the root mean square error (RMSE) of the splitting scheme compared to the analytical solutions of both test cases, for various time steps and spatial discretization lengths. Based on the RMSE values associated with test case 1, it can be suggested that the scheme is of order 1 in time and order 2 in space, as expected from its formulation (see Sect. 2.2.2). This is however less visible in test case 2, probably as a result of the RMSE varying over a smaller range of time steps and spatial discretizations lengths as in the first case. In all cases however, the scheme RMSE remains within acceptable bounds. As can be observed in Fig. 13, the numerical scheme is also able to satisfactorily reproduce the strong fluctuations of the temperature profile in both test cases, except for the minima and maxima which are truncated.

*Author contributions.* AG re-wrote and enhanced *StreamFlow* based on the original code developed by FC, performed the analysis, produced the figures and wrote the manuscript. MB helped designing the structure of *StreamFlow*, wrote the CMake scripts to compile the code and set up the CTest environment. He also provided much appreciated guidance on *MeteoIO* usage and various aspects of C++ coding. TB installed the two intermediate gauging stations in the Dischma catchment, performed the salt dilution gaugings and helped a lot in setting up the *Alpine3D* simulation. FC gave much help regarding the structure and usage of the original version of *StreamFlow* and suggested some of the analysis presented in this work. All co-authors helped write the manuscript, and HH and ML co-supervised the work.

*Acknowledgements.* This work was financially supported by the Swiss Federal Office for the Environment (FOEN), which is also greatly acknowledged for the free access to its hydrological data. All plots have been produced with the Matplotlib Python library (Hunter, 2007).





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

**Figure 1.** Schematic representation of the work flow in *StreamFlow*. Note that the first two steps are not performed in *StreamFlow* itself but in *Alpine3D* and with the help of TauDEM, respectively.





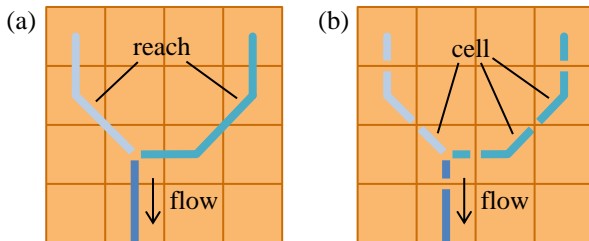

**Figure 2.** Available methods for spatially discretizing the stream reaches in *StreamFlow*: (a) the lumped approach, treating each stream reach as a lumped entity, and (b) the discretized approach, subdividing each reach into smaller entities called *cells*. Each stream reach is represented using a different shade of blue in the figure. The grid shown in brown corresponds to the DEM used by TauDEM to identify the subwatersheds and the stream network.





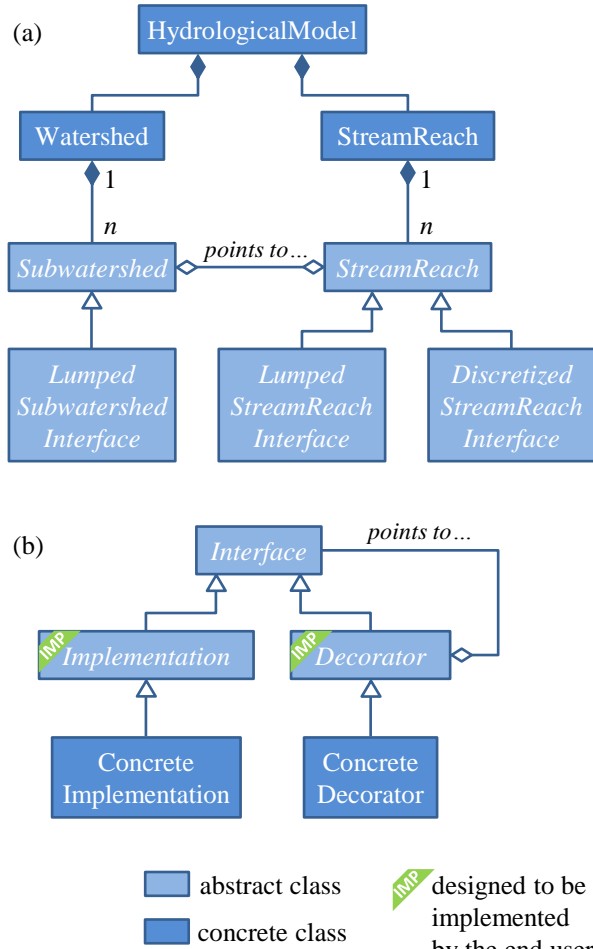

**Figure 3.** Structure of *StreamFlow*'s source code. (a) Simplified diagram of *StreamFlow*'s high level classes; (b) Diagram of the Decorator pattern used to implement abstract classes *LumpedSubwatershedInterface*, *LumpedStreamReachInterface* and *DiscretizedStreamReachInterface*.



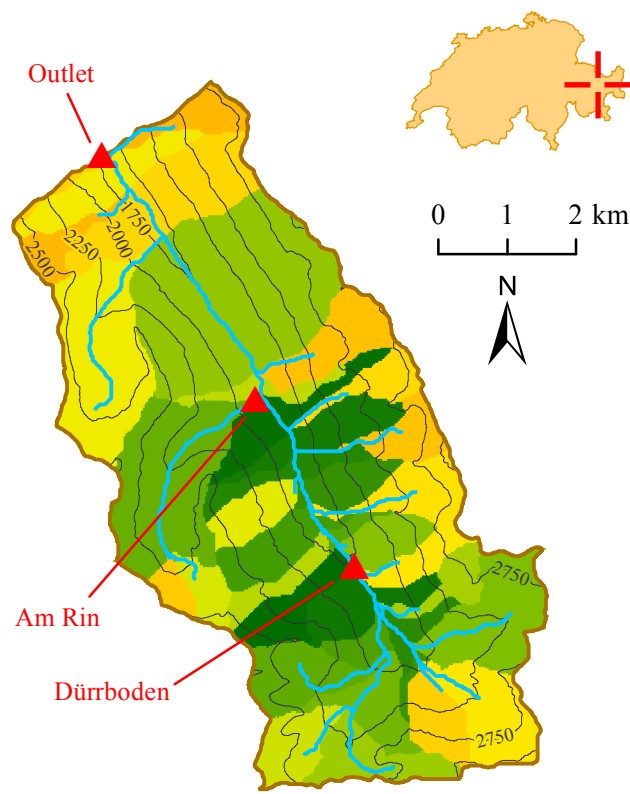

**Figure 4.** Map of the Dischma catchment displaying the subwatersheds (colored areas) and stream network (light blue line) derived from the DEM using TauDEM. The locations of the stream gauges are indicated as red triangles.

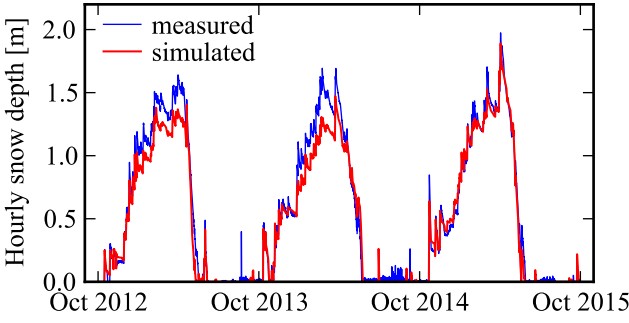

**Figure 5.** Comparison between the measured (blue line) and simulated (red line) time evolution of snow depth at the Stillberg meteorological station. The simulated curve corresponds to the mean snow depth as computed by *Alpine3D* over the $100 \times 100$ m grid cell containing the Stillberg station.





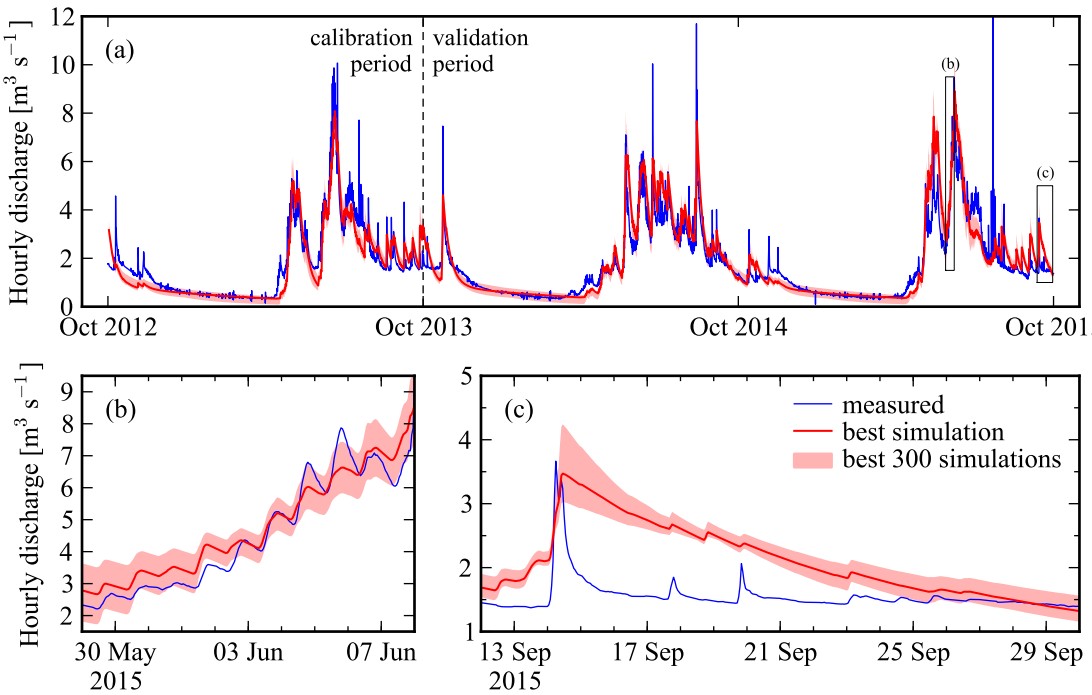

**Figure 6.** Comparison between the measured (blue line) and simulated (red line) time evolution of hourly mean discharge at the watershed outlet. Panel (a) pictures the entire simulated period, and panels (b) and (c) correspond to zooms on two selected time periods (their extents are indicated as black rectangles in panel (a)). The simulated curve was obtained with *StreamFlow* configured so as to advect water in the stream channels using the instantaneous routing approach. The uncertainty range corresponds to the 300 best runs of the model out of the 10 000 Monte-Carlo simulations.





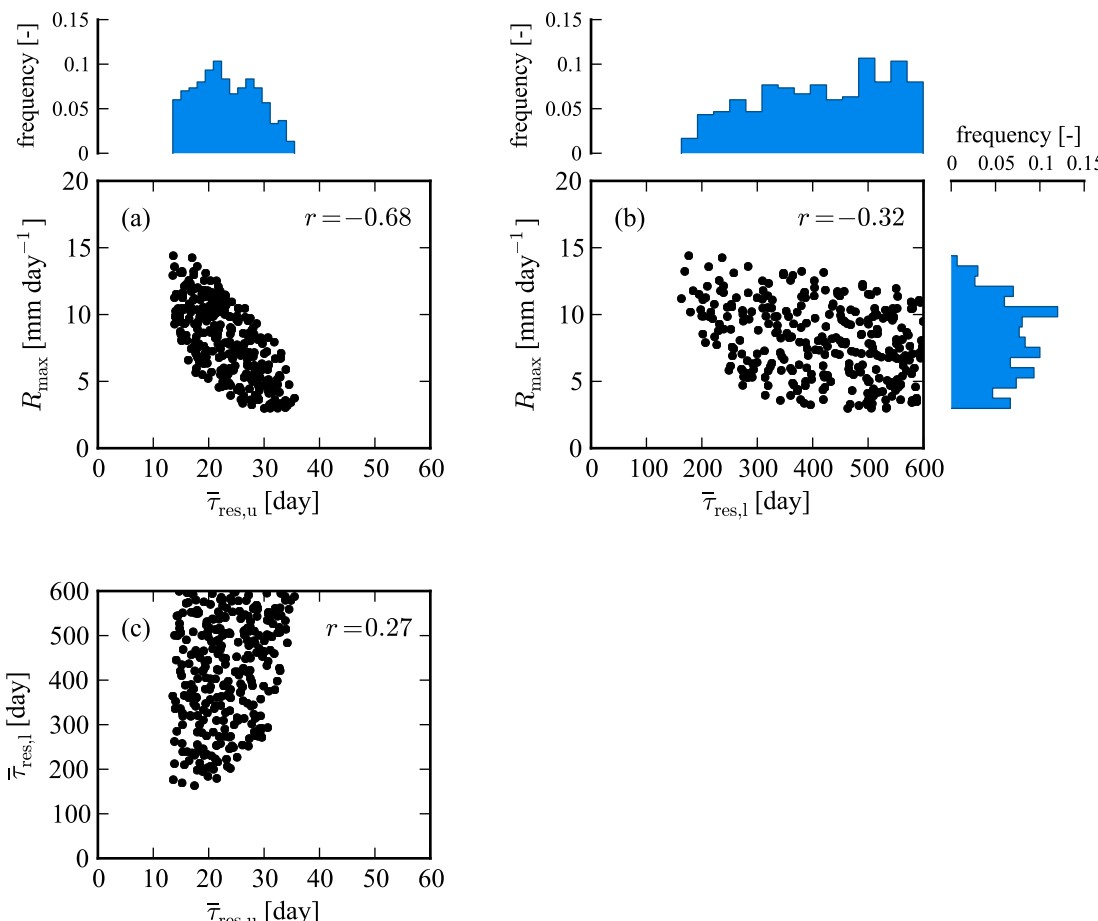

**Figure 7.** The 300 best sets of *StreamFlow* parameters associated with water transport (see Table 2 for more information on the parameters). Each panel contains the values of two parameters displayed as a function of each other: (a) $R_{max}$ versus $\overline{\tau}_{res,l}$, (b) $\overline{\tau}_{res,u}$ versus $\overline{\tau}_{res,l}$ and (c) $R_{max}$ versus $\overline{\tau}_{res,u}$. Each x or y axis spans the entire calibration range of its associated parameter. The parameter distributions are indicated in blue on the sides of the corresponding panels; for example, the distribution of the 300 best $R_{max}$ values is shown on the right-hand side of panel (a). Person's correlation coefficient $r$ between each pair of parameters is indicated in the upper right-hand corner of the associated graph.



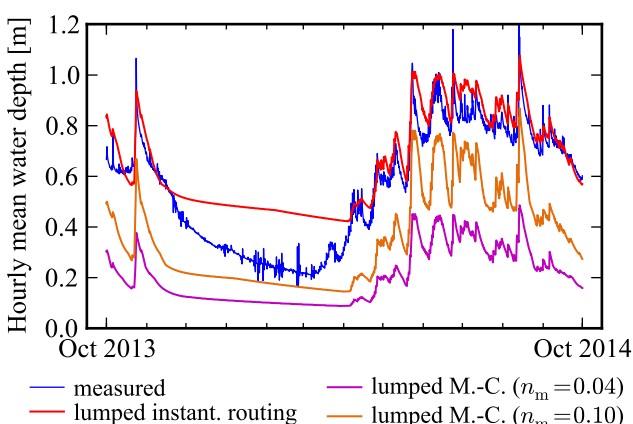

**Figure 8.** Water depth as simulated by *StreamFlow* in hydrological year 2014 using various channel water routing techniques. The measured water depth is indicated in blue and shown here only as an indication (see text). Regarding the curves associated with the Muskingum-Cunge routing technique, only those obtained using lumped stream reaches are shown (orange and violet curves). Those corresponding to discretized stream reaches almost overlap with their lumped counterparts, with the difference between each pair of curves amounting to a RMSE of 0.5 mm for $n_m = 0.04$ and 3.9 mm for $n_m = 0.10$ over the entire period 2013–2015.



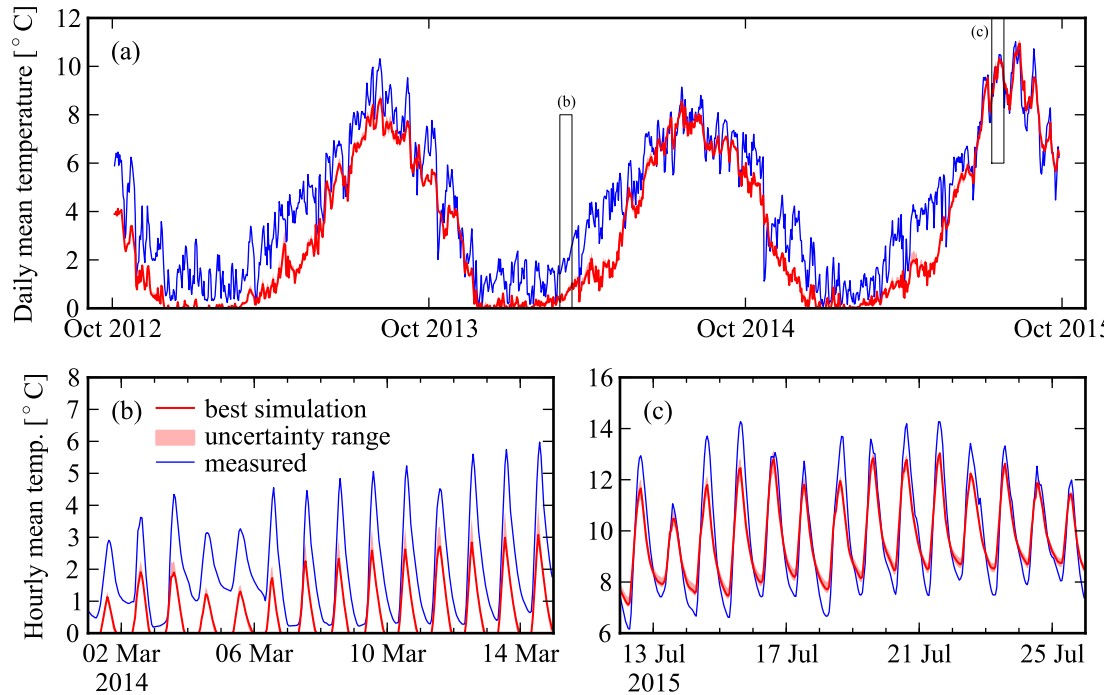

**Figure 9.** Comparison between the simulated (red line) and measured (blue line) time evolution of stream temperature at the catchment outlet. Panel (a) pictures the entire simulated period (hydrological years 2013 to 2015), with temperature aggregated into daily mean values for visibility. Panels (b) and (c) display the hourly mean temperature over two selected periods of 14 days (their respective extents are indicated as black rectangles in panel (a)). The simulated curve was obtained with *StreamFlow* based on the instantaneous water routing scheme, with the temperature of subsurface runoff being approximated as the soil temperature averaged over a depth of 2.40 m. The uncertainty range (displayed in light red) is obtained by evaluating *StreamFlow* for each one of the 300 best sets of parameters $R_{\max}$, $\overline{\tau}_{res,u}$ and $\overline{\tau}_{res,u}$ identified during calibration step 1 (see Sect. 4.2).





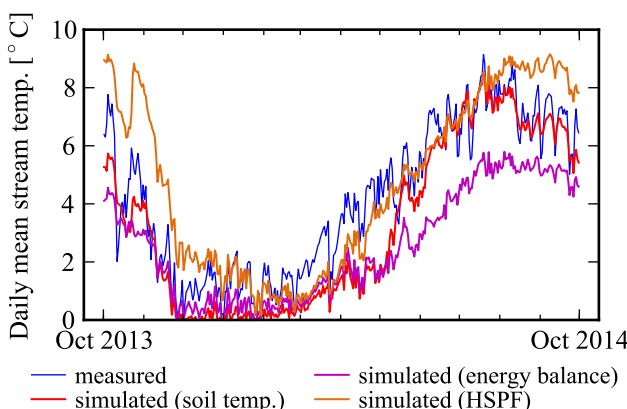

**Figure 10.** Comparison between various predictions of stream temperature at the catchment outlet, where the temperature of subsurface runoff is computed based on the following methods: the original scheme implemented in *StreamFlow* ("energy-balance"), the technique of the Hydrological Simulation Program–Fortran ("HSPF"), or as soil temperature averaged over a depth of 2.40 m ("soil temp."). All curves are aggregated into daily mean values for visibility.





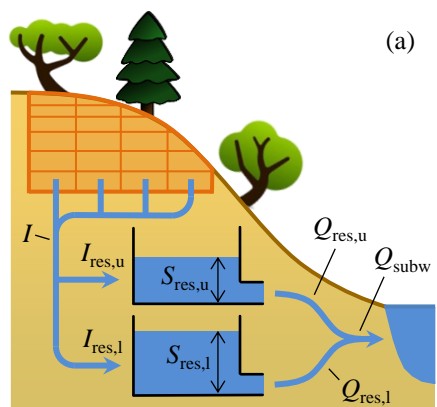

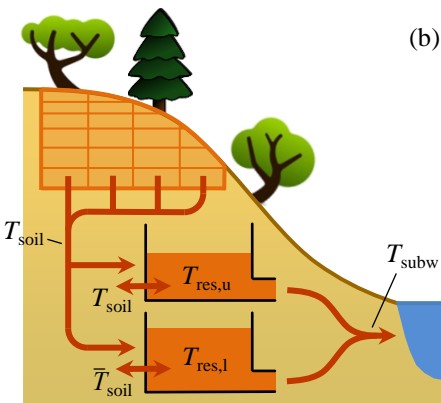

**Figure 11.** Illustrations of the models devised by Comola et al. (2015) for the computation of (a) subsurface runoff discharge, and (b) subsurface runoff temperature. The symbols are defined in the text.





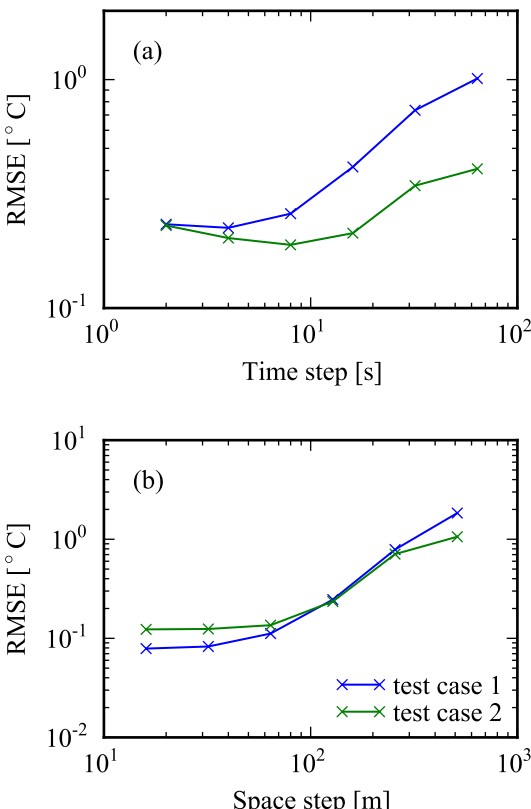

**Figure 12.** Root mean square error (RMSE) of the splitting scheme used to solve the heat balance equation in test cases 1 (blue) and 2 (green). The RMSE is computed by comparing the simulated and analytical temperature profiles at the end of the simulation (8 hours). (a) Splitting scheme RMSE for various time steps with a fixed spatial discretization length of 128 m; and (b) Splitting scheme RMSE for various spatial discretization lengths with a fixed time step of 1 s.





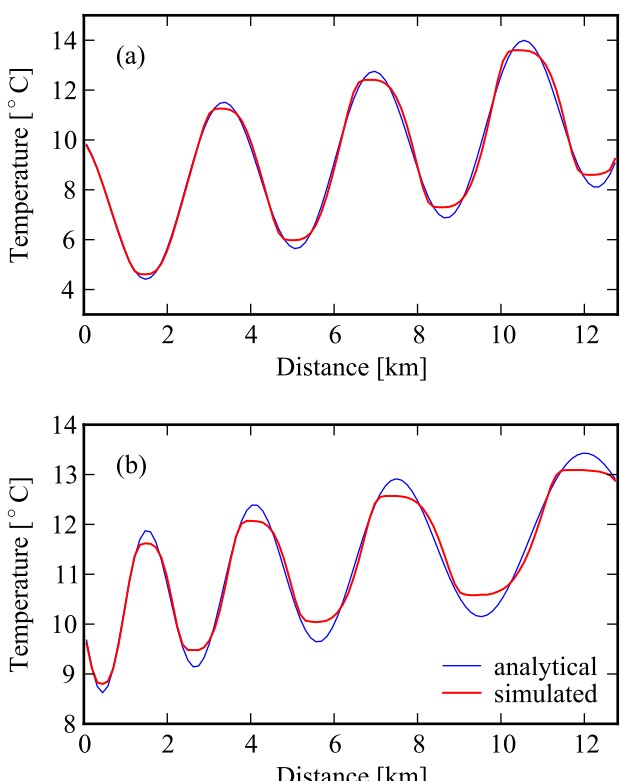

**Figure 13.** Stream temperature profile at the end of the simulation (8 hours) in (a) test case 1, and (b) test case 2. The analytical temperature profiles are displayed in blue, and those simulated by the splitting scheme in red.





**Table 1.** List of semi-distributed hydrological models which simulate both stream discharge and stream temperature and have been reviewed in the context of the present study.

| Model name | Publication | Time resolution | Target geographic location |
| --- | --- | --- | --- |
| LARSIM-WT | Haag and Luce (2008) | hourly, daily | small to large river basins |
| MODEL-Y | Sullivan et al. (1990) | hourly | forested catchments |
| SHADE-HSPF | Chen et al. (1998) | hourly | forested catchments |
| VIC-RMB | van Vliet et al. (2012) | daily | large river basins |
| CEQUEAU | St-Hilaire et al. (2000) | hourly, daily | forested catchments in Canada |
| UBC | Morrison et al. (2002) | hourly | large river basins |
| GISS GCM | Ferrari et al. (2007) | monthly | large river basins |
| SWAT | Ficklin et al. (2012) | daily, monthly | medium to large scale catchments |
| MIKE-SHE MIKE11 | Loinaz et al. (2013) | hourly | medium-scale catchments |
| WEAP21-RTEMP | Null et al. (2013) | weekly | large river basins |
| DHSVM | Sun et al. (2015) | hourly | small forested or urban catchments |
| GENESYS | MacDonald et al. (2014) | hourly | mountainous catchments |
| PCR-GLOBWB | van Beek et al. (2012) | daily | large river basins |



**Table 2.** Parameters used by *StreamFlow* to simulated water depth, discharge and temperature using various approaches. The parameters are described into more detail in the main text (Sect. 2 and Appendix A). First column of the table mentions the part of the model in which the parameter is used. The absence of a calibration range (marked as n/a) indicates a fixed parameter.

| Model part | Parameter | Units | Defined in | Calibrated or chosen value | Calibration range | Rationale for the chosen value or calibration range |
|---|---|---|---|---|---|---|
| Subwatershed outflow discharge (Sect. 2.1.1) | $R_{\max}$ | $(\mathrm{mm\,day^{-1}})$ | main text | 6.93 | $[0, 50]$ | Comola et al. (2015) |
| | $\overline{\tau}_{\mathrm{res,u}}$ | (day) | Eq. (A5) | 22.5 | $[0, 60]$ | Comola et al. (2015) |
| | $\overline{\tau}_{\mathrm{res,l}}$ | (day) | Eq. (A6) | 567.1 | $[0, 600]$ | Comola et al. (2015) |
| Subwatershed outflow temperature (Sect. 2.1.2) | $k_{\mathrm{soil}}$ | (day) | Eqs. (A7)–(A8) | 49.6 | $[0, 50]$ | Comola et al. (2015) |
| | $\tau_{\mathrm{HSPF}}$ | (day) | Eq. (1) | 58.2 | $[0.1, 100]$ | |
| | $D_{\mathrm{HSPF}}$ | (°C) | Eq. (1) | 0.99 | $[-3, 1]$ | |
| | $z_{\mathrm{d}}$ | (m) | main text | 2.40 | n/a | |
| Channel water discharge (Sect. 2.2.1) | $a_w$ | $(\mathrm{m^{-1}})$ | main text | $1.52 \times 10^{-7}$ | n/a | aerial photographs |
| | $b_w$ | (m) | main text | 0.39 | n/a | aerial photographs |
| | $\alpha_h$ | $(\mathrm{m^{1-3\beta_h}\,s^{\beta_h}})$ | main text | 0.57 | n/a | discharge gauging curve at watershed outlet |
| | $\beta_h$ | $(-)$ | main text | 0.32 | n/a | same as for $\alpha_h$ |
| | $n_{\mathrm{m}}$ | $(-)$ | Eqs. (5) and (7) | 0.04, 0.07, 0.10 | n/a | Phillips and Tadayon (2006) |
| Channel water temperature (Sect. 2.2.2) | $a_{vw}$ | $(-)$ | Eq. (13) | $2.20 \times 10^{-3}$ | n/a | Webb and Zhang (1997) |
| | $b_{vw}$ | $(\mathrm{m\,s^{-1}})$ | Eq. (13) | $2.08 \times 10^{-3}$ | n/a | Webb and Zhang (1997) |
| | $k_{\mathrm{bed}}$ | $(\mathrm{W\,m^{-2}\,K^{-1}})$ | Eq. (12) | 52.0 | n/a | Moore et al. (2005) and MacDonald et al. (2014) |

**Table 3.** Comparison of the total volume of water $V_{\mathrm{in,simu}}$ simulated by *Alpine3D* to percolate at the bottom of the watershed soil columns over each year, and the measured total volume of water $V_{\mathrm{out,meas}}$ flowing out of the catchment each year via the river.

| Hydrological year | $V_{\mathrm{in,simu}}$ $(\mathrm{m^3})$ | $V_{\mathrm{out,meas}}$ $(\mathrm{m^3})$ | Relative difference (%) |
|---|---|---|---|
| 2013 | $5.28 \times 10^7$ | $5.64 \times 10^7$ | $-6.3$ |
| 2014 | $5.88 \times 10^7$ | $5.57 \times 10^7$ | 5.7 |
| 2015 | $5.57 \times 10^7$ | $5.18 \times 10^7$ | 7.6 |





**Table 4.** Accuracy of the hourly discharge simulations performed by *StreamFlow* using the instantaneous water routing technique. The performance of the discharge benchmark model is indicated in the last row for comparison. The third column contains the period over which the error measures are computed. NSE-log corresponds to the Nash-Sutcliffe efficiency computed with the logarithm of the discharge values.

| Model | Location | Time period | RMSE $(\mathrm{m^3\,s^{-1}})$ | NSE $(-)$ | NSE-log $(-)$ | Bias $(\mathrm{m^3\,s^{-1}})$ |
|---|---|---|---|---|---|---|
| *StreamFlow* | Outlet | entire validation period | 0.60 | 0.82 | 0.90 | 0.14 |
| | Dürrboden | 17 Jan. to 25 Sept. 2015 | 0.30 | 0.81 | 0.91 | 0.11 |
| | Am Rin | 17 Jan. to 17 Jul. 2015 | 0.10 | 0.82 | 0.76 | 0.02 |
| Benchmark | Outlet | entire validation period | 0.73 | 0.74 | 0.88 | −0.04 |

**Table 5.** Accuracy of the hourly stream temperature predictions of *StreamFlow* (with $z_\mathrm{d} = 2.40$m), based on various approaches for advecting water in the stream channels and computing the temperature of subsurface runoff. The accuracy of the temperature benchmark model at the catchment outlet is indicated in the last row for comparison. All error measures are computed over the entire validation period (1st October 2013 to 1st October 2015), except at points Am Rin and Dürrboden for which the considered time period is the same as in Table 4.

| Model | Channel water routing scheme[a] | Subwatershed outflow temperature scheme[b] | Location | RMSE (°C) | NSE (−) | Bias (°C) |
|---|---|---|---|---|---|---|
| *StreamFlow* | Instantantaneous advection (lumped) | Soil temperature | Outlet | 1.45 | 0.78 | −0.88 |
| | | | Dürrboden | 1.45 | 0.78 | 0.75 |
| | | | Am Rin | 1.11 | 0.89 | −0.05 |
| | Instantaneous advection (discr.) | Soil temperature | Outlet | 1.40 | 0.80 | −0.85 |
| | Muskingum-Cunge ($n_\mathrm{m} = 0.07$, lumped) | | | 1.49 | 0.77 | −0.85 |
| | Muskingum-Cunge ($n_\mathrm{m} = 0.07$, discr.) | | | 1.46 | 0.78 | −0.80 |
| | Instantaneous advection (lumped) | Energy-balance | Outlet | 2.06 | 0.56 | −1.63 |
| | | HSPF | | 1.69 | 0.70 | 0.54 |
| Benchmark | — | — | Outlet | 1.14 | 0.87 | −0.03 |

[a] The indications "lumped" and "discr." between brackets refer to the spatial discretization of the stream reaches (see Sect. 2.2)

[b] The schemes described in Sect. 2.1.2 for the computation of subsurface runoff temperature are denoted as follows here: "soil temperature" for the scheme assuming subsurface runoff to be in thermal equilibrium with surrounding soil, "energy-balance" for the orginial scheme implemented in *StreamFlow*, and "HSPF" for the scheme inspired from the Hydrological Simulation Program–Fortran.





**Table 6.** Values chosen for the parameters of test cases 1 and 2, used to validate the numerical splitting scheme presented in Sect. 2.2.2.

| Name | Units | Value |
|------|-------|-------|
| $\tau$ | (s) | $2 \times 10^6$ |
| $\omega$ | (s$^{-1}$) | $2\pi/3600$ |
| $a_\sigma$ | (Ks$^{-1}$) | $5 \times 10^{-3}$ |
| $b_\sigma$ | (Ks$^{-1}$) | $2 \times 10^{-4}$ |
| $a_v$ | (s$^{-1}$) | $1/12800$ |
| $b_v$ | (ms$^{-1}$) | $0.5$ |
| $a_\mathrm{in}$ | (K) | $1.5$ |
| $b_\mathrm{in}$ | (K) | $283.15$ |