# Peer review of "StreamFlow 1.0: An extension to the spatially distributed snow model Alpine3D for hydrological modeling and deterministic stream temperature prediction"

_Geoscientific Model Development, 2016_

## Referee Comment (RC1) · Anonymous Referee #1 · 20 Sep 2016

The manuscript by Gallice et al. presents a revised and improved version of a hydrology and stream temperature model (StreamFlow). The authors have rewritten the source code to facilitate modularity and future development. A case study using data from a high elevation catchment in Switzerland is used to illustrate an application of the model.

This is a very nice addition to the stream temperature modelling literature and I have no substantive issues with the manuscript. I would like to make a few suggestions to help improve the clarity of the manuscript and clarify some of the model details/limitations.

Section 3.1 and Figure 3:

I struggled with some of the terminology in this section, such as Decorator and abstract vs concrete class. To ensure wide adoption of this model by scientists and practitioners who may not have extensive programming experience, I would recommend the authors try and better guide the reader through this section and limit some of the programming jargon.

Model structure and details:

I think some of the model limitations need to be presented earlier in the manuscript. For example, it's not until page 9 that the reader is made aware that the current model cannot account for riparian vegetation. In addition, it might be useful to expand on other settings where this model may not be appropriate, such as streams influenced by hyporheic exchange and deep groundwater contributions.

It also seems that a stream temperature model built for snow dominated catchments located above the tree line should be able to account for stream channel ice formation and snow inputs. Could the authors comment on why these processes were not included in the model?

Does the two linear reservoir structure from Comola et al. (2015) have a physical basis (i.e., is it meant to represent shallow subsurface flow and a deep groundwater contribution)? I realize this sort of hillslope runoff structure is common in hydrology models, but can it be appropriately extended for heat dynamics? Perhaps these details are outlined in Comola et al. (2015), but it would be nice to include more rationale on this part of the model structure in this manuscript.

Could some computation time measures be given for the model? Obviously it will depend on application and computing resources, but some general benchmarks might be useful.

Case study:

I understand that the aim of this manuscript is not to conduct a rigorous model calibration and test, but the consistent stream temperature underprediction (Figure 9a,b) and the inability of the model to properly simulate rainfall event flows (Figure 6a,c) may dissuade some readers from using this model. Is it simply a matter of more extensive model calibration to achieve better discharge and temperature predictions, or are there some model structure limitations that prevent the model from simulating event flows and/or higher stream temperatures?

From Table 5, it's interesting that most of the modelling resulted in underprediction of stream temperature; however, temperatures at Durrboden were overpredicted and Am Rin didn't show a strong bias. Why were the outlet temperatures underpredicted, but not the other two stations?

It's a nice result showing the importance of subsurface temperature runoff on stream temperature predictions at the outlet (Figure 10). The soil temperature approach seems to do a fair job, but does result in consistent underpredictions (especially during winter). Have there been any evaluations of the Alpine3D soil temperature routine? Does it account for frozen soil processes? Are the simulated zero degree soil temperatures at 2.4 m reasonable for this site? Could you recommend to readers how best to calibrate that model component, as it seems critical for accurate stream temperature predictions.

It appears that mean water depth simulations are highly sensitive to the flow routing approach used (Figure 8). How sensitive are the stream temperature predictions to the flow routing approach?

Additional comments:

Page 2, line 5 (and elsewhere): 'e.g.,' should be used to introduce examples and should not go at the end of the sentence.

Page 3, lines 1-8: Perhaps include reference and discussion of Isaak et al. 2016. Slow climate velocities of mountain streams portend their role as refugia for cold-water biodiversity, PNAS.

**GMDD**

Page 3, lines 14-16: This sentence isn't clear (e.g., what's an emergency van) and feels out of place. Consider removing or revising.

Page 4, lines 10-14: I would argue that DHSVM-RBM (Sun et al. 2015) has and can been used in mountainous terrain and employs a more process-based snow modelling approach than a degree-day method.

Page 5, line 5: Replace 'stressed out' with 'stressed'.

Page 6, line 5-6: Please explain why this approach is not compatible with potential future alternatives for subsurface runoff discharge modelling.

Page 16, line 5: '... were installed starting 16 January 2015...' until when?

Page 17, lines 24-27: How sensitive were the discharge and temperatures simulations to this warm-up period approach? Was one year for warm-up sufficient?

Page 18, lines 6-12: Could the benchmark model approach be more clearly described? I'm not sure I understand it correctly - is the benchmark model output a vector of hourly discharge (or stream temperature) that is comprised of the mean hourly values from 2005 to 2014 (i.e., temperature for day of the year 1 and hour 1 is the mean of that day-hour combination for each of 2005 to 2014)?

Figure 5: Is this comparing snow depth or snow water equivalent? Wouldn't snow water equivalent be more appropriate?

Figure 9: The observed stream temperature doesn't suggest this stream freezes in the winter. Is that the case? Does surface ice form? If not, what energy exchange processes are maintaining stream temperature above zero degrees Celsius?

Table 2: It seems that the calibration of k_soil and D_HSPF are pushing the upper limits of the calibration range. Should a wider range have been selected?

Table 3: This table could probably be removed, since these results are already presented in the text.

[Figure]

---

## Referee Comment (RC2) · Anonymous Referee #2 · 24 Oct 2016

Overall comment

The authors have done an excellent job preparing this manuscript and in their revision of the StreamFlow model for this application. I feel that this will be a valuable contribution to the stream temperature literature and do not have any major concerns with the current version of the manuscript. However, I do suggest minor revisions to the current draft in order to better make use of this excellent work within the broader stream temperature field. I also agree with the points made by Reviewer 1; therefore, will not reiterate.

Introduction

Page 2 - Line 25 - The authors mention there is a strong correlation between stream temperature and air temperature. This is true; however, this does not always imply causation (Johnson, 2003). Recent work has demonstrated how important this fact is in terms of modelling and in understanding stream temperature response to environmental change. It would be useful to expand this discussion within the context of this particular model.

Page 3 - Paragraph 2 - I am not sure why this paragraph is here. It seems out of place.

Section 2

The use of the term "subsurface runoff" has been applied throughout. This is a strange use of the word runoff given that it typically applies to shallow or overland flow. Perhaps consider using the word "flow" rather than "runoff".

The Tsubw description is fairly vague. I imagine sub watershed temperature plays a substantial role in the overall stream energy balance, yet it is not well described. Further explanation is required.

Overall, the manuscript would greatly benefit from a more formal sensitivity analysis so that the reader can understand how each of the terms used can influence stream temperature in this model. It is not clear how each of the terms is being simulated and what their relative influence on temperature is. For example, the authors suggest that the underestimation of the diurnal temperature pattern is likely due to stream width and depth, or subsurface temperature. There is no discussion of how the influence of the radiative balance (by far the largest term). It's also not clear how hyporheic fluxes play a role. Quantifying these various fluxes and their role in governing stream temperature would be an excellent use of this modelling tool.

---

## Author Comment (AC1) · 3 Nov 2016

We wish to thank the reviewer for his/her comments and his/her interest in our work. Below are our responses to the points raised by the reviewer. Each original remark of the reviewer is indicated in bold italics and directly followed by our corresponding answer. The modifications brought to the manuscript are indicated in blue.

*"I struggled with some of the terminology in [S]ection [3.1 and Figure 3], such as Decorator and abstract vs concrete class. To ensure wide adoption of this model*

*by scientists and practitioners who may not have extensive programming expe-*
*rience, I would recommend the authors try and better guide the reader through*
*this section and limit some of the programming jargon."*

We acknowledge that the original formulation was targeted more at programmers than
at scientists. Following the reviewer's suggestions, we completely rewrote the para-
graph using less jargon and better explaining the terms of jargon which we kept. We
hope that the new wording (see below) is more understandable for people without a
strong programming background.

[revised manuscript text omitted]

*"I think some of the model limitations need to be presented earlier in the manuscript. For example, it's not until page 9 that the reader is made aware that the current model cannot account for riparian vegetation. In addition, it might be useful to expand on other settings where this model may not be appropriate, such as streams influenced by hyporheic exchange and deep groundwater contributions."*

We agree that the model limitations were not particularly emphasized in the original manuscript. We therefore inserted the following sentence at the end of the introductory section to make clear that the present version of the model does not account for riparian shading: "In its present form, the model application is restricted to catchments located above the tree line or with little to no vegetation cover along the stream, due to the absence of a proper riparian vegetation module." We also added a sentence in Sect. 2.2.1 to emphasize the fact that hyporheic exchanges are neglected, but could theoretically be taken into account by appropriately choosing the values of $k_{bed}$ and $T_{bed}$. Regarding the "deep groundwater contributions" mentioned by the reviewer, we actually believe that the model is able to account for them, given that one relies on the HSPF approach or the original formulation of Comola et al. (2015) to compute the temperature of subsurface runoff. As a matter of fact, both of these approaches are able to simulate an approximately constant subsurface runoff temperature over the year, given that the model parameters are chosen suitably.

*"It also seems that a stream temperature model built for snow dominated catchments located above the tree line should be able to account for stream channel ice formation and snow inputs. Could the authors comment on why these processes were not included in the model?"*

We are not really sure what the reviewer means with "snow inputs." In case (s)he refers to lateral water inflow into the stream originating from snow melt, we would like to mention that the model does actually take it into account: the snow melting and

water percolation into the soil are modeled in *Alpine3D*. Regarding the "channel ice formation", we certainly acknowledge that the model misses an important component. This phenomenon is however extremely hard to simulate in small Alpine streams, due to the high turbulence of the flow and the presence of very large emerged boulders in the stream bed. Field observations in the Dischma catchment revealed that ice formed at some places along the stream, but not in others. In the absence of any substantial knowledge on this phenomenon, we preferred not to attempt at modeling it for the moment – although we agree with the reviewer that this point should definitely be improved upon in the future.

*"Does the two linear reservoir structure from Comola et al. (2015) have a physical basis (i.e., is it meant to represent shallow subsurface flow and a deep groundwater contribution)? I realize this sort of hillslope runoff structure is common in hydrology models, but can it be appropriately extended for heat dynamics? Perhaps these details are outlined in Comola et al. (2015), but it would be nice to include more rationale on this part of the model structure in this manuscript."*

The structure from Comola et al. (2015) has some sort of physical basis, although the attribution of one type of flow (shallow subsurface or groundwater) to each one of the two reservoirs is not particularly evident. The picture is complicated by the fact that both reservoirs are fed by the percolating water flux at a depth of 1 m, so that the upper reservoir cannot account for shallow subsurface flow. On the other hand, it is true that the upper reservoir is intended to represent a faster component of the subwatershed response as compared to the lower reservoir. Comola et al. (2015) have shown that this structure could be successfully used to model stream temperature in high Alpine watersheds. We agree with the reviewer that adding more details on the structure from Comola et al. (2015) in our paper would make it more self-contained, however we believe that the current manuscript is already very long. We therefore prefer referring the interested reader to the original paper, briefly summarizing the main equations in

Appendix A.

*"Could some computation time measures be given for the model? Obviously it will depend on application and computing resources, but some general benchmarks might be useful."*

We thank the reviewer for this interesting suggestion. We added some rough benchmarks at the beginning of Sect. 3 (see lines below).

"Regarding the computation time of *StreamFlow* itself, we observed the simulation duration to be highly dependent on the methods used to compute the transport of water and heat along the stream network. In general, the lumped approaches are associated with much reduced computation times as compared to the discretized methods, and the Muskingum-Cunge water routing technique is slower than its instantaneous advection counterpart. As an indication, the stream temperature simulations reported in Sect. 4, which were run on a personal computer with 2 GB of RAM and an Intel® Core™ i7 processor, took between a few seconds (with the lumped instantaneous routing approach) and more than 24 h (with the discretized Muskingum-Cunge approach) to complete."

*"I understand that the aim of this manuscript is not to conduct a rigorous model calibration and test, but the consistent stream temperature underprediction (Figure 9a,b) and the inability of the model to properly simulate rainfall event flows (Figure 6a,c) may dissuade some readers from using this model. Is it simply a matter of more extensive model calibration to achieve better discharge and temperature predictions, or are there some model structure limitations that prevent the model from simulating event flows and/or higher stream temperatures?"*

We thank the reviewer for his/her remark. Regarding the discharge simulations, the very poor modeling of short-lived discharge peaks is actually a question of model limitation, since *StreamFlow* does not feature any overland flow and/or shallow subsurface flow module. This absence is due to the fact that *Alpine3D* is only able to simulate vertical water flows at the moment. The simulation of overland flow would imply some part of the vertical water flux computed in *Alpine3D* to be converted into a horizontal flux, which would require substantial work at the very core of *Alpine3D*. Alternatively, *StreamFlow* could be provided with grids of the vertical water flux computed by *Alpine3D* at different depths in the soil, and use them to compute the various components of the subwatershed outlet discharge (i.e. surface runoff, shallow subsurface flow and groundwater). Future versions of the code might correct this limitation, but nothing is unfortunately planned in the near future. We added a few words in the manuscript to summarize this: "This model limitation could be fixed by implementing a new method for transferring water across the subwatersheds, based on a more physically-based approach than the linear reservoir method used here. Such an improvement is not planned soon, but might become available in a future version of *StreamFlow*."

Regarding the stream temperature simulations, we believe that the general underestimation of modeled temperature is mostly due to the poor vertical resolution of the soil columns in *Alpine3D*. As a matter of fact, this poor resolution is expected to turn into in the underestimation of soil temperature and hereby subsurface runoff temperature, which ultimately results into the underestimation of stream temperature. We inserted a few sentences in the manuscript to clarify this point and suggest some workaround: "As evident from panel (a), stream temperature is generally underestimated by the model on a daily time scale, particularly during the snow melt season in spring. We attribute this discrepancy to soil temperature as simulated by *Alpine3D* being too low, since its value averaged down to $2.40$ m typically remains around $0\,°$C until mid-June (not shown). This underestimation of soil temperature is in turn expected to result from the coarse soil vertical discretization used in the *Alpine3D* simulation (see above). As a workaround, the soil temperature averaging depth $z_\mathrm{d}$ could be increased, since it would result in larger soil temperature values while at the same time not lengthening the computation time of the *Alpine3D* simulation. This approach is not presented here

into more detail in order to keep the article concise, but simply mentioned as a hint to interested readers."

*"From Table 5, it's interesting that most of the modelling resulted in underprediction of stream temperature; however, temperatures at Durrboden were overpredicted and Am Rin didn't show a strong bias. Why were the outlet temperatures underpredicted, but not the other two stations?"*

There are two reasons explaining this difference:

1. The time period over which the simulations are compared with the observations is not the same in all three cases: the model predictions at Dürrboden and Am Rin are only evaluated over the period January–September 2015 and January-July 2015, respectively. As such, the evaluation period at these two locations does not include all of autumn/winter, which is exactly the time period during which the model underestimates water temperature.

2. Water temperatures measured at Dürrboden and Am Rin are much closer to the simulated soil temperature as compared to the water temperature measured at the catchment outlet. As a consequence, the matching between measured and simulated water temperature at the two intermediate points is much better.

We added some sentences in the manuscript to clarify these two points: "Regarding the model performance at the two intermediate gauging points, the values of the error measures at Dürrboden are found to be essentially the same as at the outlet point, except for the positive bias (see Table 5). The latter indicates that the modeled stream temperatures at these two points are – contrary to the temperature at the outlet point – not underestimated, which results from the fact that the stream temperature values measured at the intermediate points are much closer to the simulated soil temperature curve in spring as compared to the stream temperature measured at outlet point (not

Interactive
comment
shown). Concerning Am Rin, the apparent better values for RMSE, NSE and bias have to be weighted against the short time period over which they are evaluated (17 January 2015 to 17 July 2015)."

*"It's a nice result showing the importance of subsurface temperature runoff on stream temperature predictions at the outlet (Figure 10). The soil temperature approach seems to do a fair job, but does result in consistent underpredictions (especially during winter). Have there been any evaluations of the Alpine3D soil temperature routine? Does it account for frozen soil processes? Are the simulated zero degree soil temperatures at 2.4 m reasonable for this site? Could you recommend to readers how best to calibrate that model component, as it seems critical for accurate stream temperature predictions."*

The *Alpine3D* soil temperature routine does indeed account for frozen soil processes and has been evaluated in several papers (Luetschg et al., 2004; Luetschg and Haeberli, 2005). However, we agree with the reviewer that the simulated depth-averaged soil temperature is far too low for the chosen site. As mentioned above, we believe this temperature underestimation to result from the coarse vertical resolution of the soil columns used in the *Alpine3D* simulation. Following the reviewer's advice, we added some suggestions on how to improve the accuracy of the soil temperature simulations (see above).

*"It appears that mean water depth simulations are highly sensitive to the flow routing approach used (Figure 8). How sensitive are the stream temperature predictions to the flow routing approach?"*

As pointed out by the reviewer, the water depth predictions vary noticeably depending on the flow routing technique used. This is nevertheless not the case for simulated stream temperature, as mentioned in ll. 32–35 p. 35 and ll. 1–5 p. 32 of the original

manuscript (see also Table 5).

*"Page 2, line 5 (and elsewhere): 'e.g.,' should be used to introduce examples and should not go at the end of the sentence."*

We thank the reviewer for his/her observation and corrected the manuscript accordingly.

*"Page 3, lines 1–8: Perhaps include reference and discussion of Isaak et al. 2016. Slow climate velocities of mountain streams portend their role as refugia for cold-water biodiversity, PNAS."*

We would like to thank the reviewer for suggesting a discussion of this particularly interesting article, which we did not know about. We added the following sentence at the end of the paragraph: "In addition, a recent study by Isaak et al. (2016) points at the fact that mountain streams may be more resilient to warming air temperatures as was considered until now, implying that they may play an important role in the preservation of cold-water species in the future."

*"Page 3, lines 14–16: This sentence isn't clear (e.g., what's an emergency van) and feels out of place. Consider removing or revising."*

Following the recommendations of the second reviewer, we actually removed the entire paragraph.

*"Page 4, lines 10–14: I would argue that DHSVM-RBM (Sun et al. 2015) has and can been used in mountainous terrain and employs a more process-based snow modelling approach than a degree-day method."*

We thank the reviewer for his/her remark and modified the manuscript accordingly, i.e. we removed the citation of the article referring to DHSVM-RBM from the list of models "being used over low-altitude catchments," and specified that "the model of Sun et al. (2015) [. . . ] has been tested over Alpine watersheds." We however did not modify the last sentence of the paragraph, since we only state that "most" (and not *all*) of the models "rely[. . . ] on the degree-day method," hereby implicitly recognizing that some of the models are based on more advanced techniques.

*"Page 5, line 5: Replace 'stressed out' with 'stressed'."*

We followed the reviewer's recommendation.

*"Page 6, line 5–6: Please explain why this approach is not compatible with potential future alternatives for subsurface runoff discharge modelling."*

We split and rephrased the sentence in the hope that it is now clearer: "The first approach corresponds to the one developed by Comola et al. (2015), which performs a simplified energy-balance of subsurface water at the subwatershed scale. Since this method specifically requires the subwatershed outlet discharge to be modeled exactly as in Sect. 2.1.1, it is not compatible with potential future alternatives for modeling the subsurface flux discharge."

*"Page 16, line 5: '. . . were installed starting 16 January 2015. . . ' until when?"*

We modified the main text to include the dates at which the stations were removed (these dates were already present in Table 4 and in the caption of Table 5).

*"Page 17, lines 24–27: How sensitive were the discharge and temperatures simulations to this warm-up period approach? Was one year for warm-up sufficient?"*

[Figure]

The simulations were quite sensitive to the warm-up period, especially the discharge ones. In the absence of warm-up, the percolating soil water is almost entirely used to fill up the subwatershed reservoirs during the first three to four months. As a consequence, the watershed outlet discharge is generally underestimated during this time period. Only a few tests were run, but it seems that one year of warm-up is sufficient to let the reservoir levels adapt to the inflow conditions. We added a sentence in the manuscript to better explain the purpose of the warm-up period: "This approach is observed to improve the quality of the simulation – notably of modeled discharge – by letting enough time for the amount of water stored in the linear reservoirs representing the subwatersheds to adapt to the inflow conditions (not shown)."

*"Page 18, lines 6–12: Could the benchmark model approach be more clearly described? I'm not sure I understand it correctly – is the benchmark model output a vector of hourly discharge (or stream temperature) that is comprised of the mean hourly values from 2005 to 2014 (i.e., temperature for day of the year 1 and hour 1 is the mean of that day–hour combination for each of 2005 to 2014)?"*

We recognize that the original description of the benchmark model was not necessarily obvious, although the reviewer did understand correctly what we meant. We hope that the new wording is clearer: "Two benchmark models are actually considered here, one for discharge and one for temperature. Both are constructed by averaging, for each hour of each day of the year, the values of discharge and temperature measured at the catchment outlet on those particular hour and day between 2005 and 2014. For example, the output of the discharge benchmark model on 1$^{st}$ January at 1 pm is the same for all years and corresponds to the average of the ten discharge values measured at the catchment outlet on 1$^{st}$ January at 1 pm from 2005 to 2014."

*"Figure 5: Is this comparing snow depth or snow water equivalent? Wouldn't snow water equivalent be more appropriate?"*

Figure 5 compares snow depth (see y-axis and caption). We fully agree with the reviewer that comparing snow water equivalent would have been more appropriate, however no continuous measurements of snow water equivalent were available in the Dischma catchment over the considered time period.

*"Figure 9: The observed stream temperature doesn't suggest this stream freezes in the winter. Is that the case? Does surface ice form? If not, what energy exchange processes are maintaining stream temperature above zero degrees Celsius?"*

As pointed out by the reviewer, the Dischma stream does not entirely freeze in winter. Surface ice forms in some places, especially along the banks, but not everywhere (see above). For example, the central part of the main stream is generally ice-free over the entire winter, and flowing water is observed at all times in the main channel year-round. We believe that this results from the stream being mostly fed by groundwater (especially in winter), whose temperature is sufficiently high to prevent freezing. The stream is also highly turbulent, hereby hindering the formation of ice. We however admit a lack of understanding of all the exact hydrological processes taking place in the Dischma catchment, since most of the field investigations conducted in this catchment to date have concentrated more on snow than on hydrology.

*"Table 2: It seems that the calibration of $k_{soil}$ and $D_{HSPF}$ are pushing the upper limits of the calibration range. Should a wider range have been selected?"*

The procedure generally followed for calibration consists in first determining a plausible range of values for each parameter based on physical considerations, and then determine the value within this range which leads to the best simulation results. Regarding parameter $k_{soil}$, we decided to take the same range of plausible values as in Comola et al. (2016) for consistency. On the other hand, the plausible range for $D_{HSPF}$ was cho-

[Figure]

sen more or less empirically, without any particular physical justification. We therefore agree with the reviewer that, based on the calibration results, the prior range of $D_{\text{HSPF}}$ should certainly have been chosen wider. We however believe that the results would not have changed significantly, and the main conclusion would certainly have been the same. We added a few sentences at the end of the first paragraph in Sect. 4.3.3 to address this point: "Similarly, it can be observed based on Table 2 that the calibrated values of some stream temperature parameters (notably $k_{\text{soil}}$ and $D_{\text{HSPF}}$) are close to the respective upper limits of their associated calibration ranges. For the sake of conciseness, we however proceed with the parameter values presented in Table 2 and postpone the in-depth evaluation of the model sensitivity with respect to its parameters to a future publication."

*"Table 3: This table could probably be removed, since these results are already presented in the text."*

We agree with the reviewer that the main results of the table are already presented and discussed in the main text. We would nevertheless wish to keep the table, since it also gives indications on the total amount of water transiting through the watershed each year, which we believe might be of interest to some readers.

---

## Author Comment (AC2) · 3 Nov 2016

We wish to thank the reviewer for his/her comments and his/her interest in our work. Below are our responses to the points raised by the reviewer. Each original remark of the reviewer is indicated in bold italics and directly followed by our corresponding answer. The modifications brought to the manuscript are indicated in blue.

*"Page 2 - Line 25 – The authors mention there is a strong correlation between stream temperature and air temperature. This is true; however, this does not*

*always imply causation (Johnson, 2003). Recent work has demonstrated how important this fact is in terms of modelling and in understanding stream temperature response to environmental change. It would be useful to expand this discussion within the context of this particular model."*

We fully agree with the reviewer that correlation does not imply causation and acknowledge that the original sentence in the manuscript was wrong. We therefore replaced "result in" with "be associated with" in the original phrase "the increase in air temperature is expected to result in globally higher stream temperatures over the year." We however believe that a further discussion of correlation and causation would not be particularly relevant in the present paper which focuses on the presentation of a deterministic model, regardless of the fact that the manuscript is – in our opinion – already very long.

*"Page 3 - Paragraph 2 – I am not sure why this paragraph is here. It seems out of place."*

Following the reviewer's advice, we entirely removed the paragraph.

*"The use of the term 'subsurface runoff' has been applied throughout. This is a strange use of the word runoff given that it typically applies to shallow or overland flow. Perhaps consider using the word 'flow' rather than 'runoff'."*

We wish to thank the reviewer for his/her remark. As non-native speakers, it indeed did not occur to us that the use of the term 'subsurface runoff' was strange. We therefore replaced all occurrences of this term with 'subsurface water flux', which actually better describes what we really mean.

*"The $T_{\mathrm{subw}}$ description is fairly vague. I imagine sub watershed temperature*

*plays a substantial role in the overall stream energy balance, yet it is not well described. Further explanation is required."*

We recognize that the original sentence in which $T_{subw}$ was defined might have lacked some clarity. The new sentence, which we hope is now clearer, reads: "In StreamFlow, the discharge $Q_{subw}$ (m$^3$ s$^{-1}$) of the subsurface water flux generated by each subwatershed is computed independently from its temperature $T_{subw}$ (K)."

*"Overall, the manuscript would greatly benefit from a more formal sensitivity analysis so that the reader can understand how each of the terms used can influence stream temperature in this model."*

We agree with the reviewer that a formal sensitivity analysis would be a nice complement to our work. However, in our opinion, this analysis should be the subject of a second paper rather than being inserted in the present manuscript. As a matter of fact, the present manuscript appears to us as already very large. In addition, its main purpose is to describe the components and structure of the model rather than to make an exhaustive assessment of the model characteristics.

*"It is not clear how each of the terms is being simulated and what their relative influence on temperature is. For example, the authors suggest that the underestimation of the diurnal temperature pattern is likely due to stream width and depth, or subsurface temperature. There is no discussion of how the influence of the radiative balance (by far the largest term). It's also not clear how hyporheic fluxes play a role. Quantifying these various fluxes and their role in governing stream temperature would be an excellent use of this modelling tool."*

We agree with the reviewer that the use of the model to quantify the various heat fluxes would be a nice application. However, as stated above, the present paper is aimed at describing the model components, and not so much at studying stream temperature

dynamics using the model. This is the reason why we decided to submit our manuscript to GMD, which specifically focuses on model descriptions. The model application does, in our opinion, not fit in the present paper and would need to be presented in a separated article.

––––––––––––––––––––––––––––––